# A new RANS-based wind farm parametrization and inflow model for wind farm cluster modeling

Maarten Paul van der Laan[1], Oscar García-Santiago[1], Mark Kelly[1], Alexander Meyer Forsting[1], Camille Dubreuil-Boisclair[2], Knut Sponheim Seim[2], Marc Imberger[1], Alfredo Peña[1], Niels Nørmark Sørensen[1], and Pierre-Elouan Réthoré[1]

[1]Technical University of Denmark, DTU Wind Energy, Risø Campus, Frederiksborgvej 399, 4000 Roskilde, Denmark
[2]Equinor ASA, Oslo, Norway

**Correspondence:** Maarten Paul van der Laan (plaa@dtu.dk)

**Abstract.** Offshore wind farms are more commonly installed in wind farm clusters, where wind farm interaction can lead to energy losses; hence, there is a need for numerical models that can properly simulate wind farm interaction. This work proposes a Reynolds-averaged Navier-Stokes (RANS) method to efficiently simulate the effect of neighboring wind farms on wind farm power and annual energy production. First, a novel steady-state atmospheric inflow is proposed and tested for the application of RANS simulations of large wind farms. Second, a RANS-based wind farm parametrization is introduced, the actuator wind farm (AWF) model, which represents the wind farm as a forest canopy and allows to use of coarser grids compared to modeling all wind turbines as actuator disks (ADs). When the horizontal resolution of the RANS-AWF model is increased, the model results approach the results of the RANS-AD model. A double wind farm case is simulated with RANS to show that replacing an upstream wind farm with an AWF model only causes a deviation less than 1% in terms of wind farm power of the downstream wind farm. Most importantly, a reduction in CPU hours of 75.1% is achieved, provided that the AWF inputs are known, namely, wind farm thrust and power coefficients. The reduction in CPU hours is further reduced when all wind farms are represented by AWF models; namely 92.3% and 99.9%, for the double wind farm case and for a wind farm cluster case consisting of three wind farms, respectively. If the wind farm thrust and power coefficient inputs are derived from RANS-AD simulations then the CPU-time reduction is still 82.7% for the wind farm cluster case. For the double wind farm case, the RANS models predict different wind speed flow fields compared to output from simulations performed with the mesoscale Weather Research and Forecasting model, but the models are in agreement with the inflow wind speed of the downstream wind farm. The RANS-AD-AWF model is also validated with measurements in terms of wind farm shape; the model captures the trend of the measurements for a wide range of wind directions, although the measurements indicate more pronounced wind farm wake shapes for certain wind directions.

## 1 Introduction

The growth of offshore wind energy has led to wind farm clustering, where wind farm interaction is unavoidable. Recently, the Danish government released a report with plans for a new 10 GW offshore wind farm cluster situated around an artificial energy island hub in the North Sea (COWI, 2020). This wind farm cluster consists of ten 1 GW wind farms with a wind farm

inter-distance of only 8 km using a non-optimized wind farm cluster layout. More examples of wind farm clusters can be found in other parts of the North Sea, the Baltic Sea and the East China Sea (4coffshore.com). To our best knowledge, there are currently no International Electrotechnical Commission (IEC) standards regarding recommended minimal distances between wind farms to avoid losses due to wind farm wakes and wind farm owners do not have any control over potential newly built neighboring wind farms. Hence, there is need for improved numerical models that can estimate wind farm wake losses (and potential gains due to speed up effects) in order to establish standards for wind farm placement in wind farm clusters.

Low-fidelity engineering wind turbine wake models (Göçmen et al., 2016) can be used to model wind farm interaction; however, large model uncertainties exist due assumptions on wake shape and wake superposition methods. The performance of these models is case dependent and can be poor (Fischereit et al., 2022) or reasonable (Nygaard et al., 2020), which often depends on how the models are calibrated. The current state-of-the-art numerical models employed to simulate wind farm clusters are based on mesoscale models, e.g., the Weather Research and Forecasting (WRF) model (Skamarock et al., 2019). The WRF model is normally run with nested domains, where the finest domain has a typical horizontal cell length of 1 km. The effect of a wind farm is modeled by a simple wind farm parametrization, where a drag force is added to the momentum equation (Fitch et al., 2012; Volker et al., 2015; Abkar and Porté-Agel, 2015), and its magnitude is based on the wind turbine thrust curve. Fitch et al. (2012) also included a source of turbulent kinetic energy based on the wind turbine extracted kinetic energy that is not converted to electricity. A turbulent kinetic energy source term was also motivated byAbkar and Porté-Agel (2015) based on dispersive Reynolds-stresses due to under resolving the wind farm induced turbulent kinetic energy. Volker et al. (2015) did not include such a source term but takes into account sub-grid-scale vertical wake expansion within one grid cell.

It is not trivial to verify the wind farm parametrizations in mesoscale models because, e.g., in the WRF model the wind farm parametrizations are implemented within 1D planetary boundary layer (PBL) schemes (Peña et al., 2022). These schemes are not scale aware and become horizontal resolution-dependent when decreasing the grid size below $\approx 1$ km. Only until recently, a new fully 3D PBL scheme was implemented in the WRF model (Kosović et al., 2020) and the Fitch parametrization was also already configured to run with it (Rybchuk et al., 2022). In this work, we propose a Reynolds-averaged Navier-Stokes (RANS) based wind farm model, the actuator wind farm model (AWF), which can be used to verify mesoscale wind farm parametrizations in a microscale environment because the minimal horizontal spacing is not limited. The AWF represents a wind farm as a forest canopy using distributed drag forces. The use of a forest canopy model to represent a single wind turbine wake has been employed by Réthoré (2009), although the model performed poorly compared to a high-fidelity turbulence resolving simulation. In the present work, we will show that the use of a forest canopy model for the entire wind farm can work quite well, as long as the correct wind farm drag force magnitude and distribution is applied. This is achieved by applying a pre-calculated wind farm drag force magnitude that is both wind speed and wind direction dependent, and by employing a horizontal drag force distribution as a superposition of two-dimensional Gaussian functions centered at the wind turbine locations. The latter represents a smoothed number of wind turbines per cell as opposed to counting the number of wind turbines per cell as commonly used in WRF's wind farm parametrization, in order to overcome aliasing effects that can lead to artificial wind farm wake shapes. It is common to use Gaussian functions to distribute forces of simplified wind turbine

models in Computational Fluids Dynamics (CFD), as performed for actuator line (Sørensen and Shen, 2002), sector (Storey et al., 2015) and disk (AD) (Mikkelsen, 2003; Wu and Porté-Agel, 2011) models. The AWF model can also be used to calculate the annual energy production of a large wind farm cluster, which would become computationally very expensive if all wind turbines are modeled by ADs that require a horizontal cell size in the order of $D/8$, where $D$ is the rotor diameter. If the effect of neighboring wind farms on a single wind farm needs to be simulated, the wind farm of interest can be modeled as ADs using a fine horizontal spacing, while all the other wind farms can be modeled as AWFs using a much coarser horizontal spacing; we refer to this model as RANS-AD-AWF. Such a numerical setup is only slightly more expensive in terms of computational costs compared to simulating the wind farm of interest without the neighboring wind farms. A further reduction in computational effort can be achieved by modeling all wind farms as AWFs, hereafter known as the RANS-AWF model.

Wind farm (cluster) modeling with RANS is challenging when an inflow model with an atmospheric boundary layer (ABL) height is employed together with a large horizontal domain in the order of 200 km and more (van der Laan et al., 2017). This is because the eddy viscosity above the ABL height is very small and numerical instabilities can appear if the domain is large enough. These numerical instabilities could be interpreted as numerical gravity waves because a solution with standing waves can be obtained when flow is solved with an unsteady RANS method. One could employ a more robust inflow model representing a neutral surface layer inflow; however, such a model is not very realistic for large wind turbines that are expected to operate outside the surface layer and neither for a wind farm cluster where the effect of the Coriolis force can become important (van der Laan and Sørensen, 2017b). An ABL inflow model can be obtained when using the global turbulence length scale limiter of Apsley and Castro (1997), since it does not require a temperature equation and the ABL height is determined by setting a maximum turbulence length scale. In addition, the model is only dependent on two non-dimensional numbers (van der Laan et al., 2020), which eases getting the desired inflow profile from a library of pre-calculated ABL profiles. However, the global turbulence length scale limiter also limits all turbulence length scales in the three-dimensional domain, as wind farm wake turbulence length scales. The latter can lead to a wind farm wake that stops recovering with downstream distance, which is a non-physical result. Furthermore, the effective inversion strength is implicit in this model, and is relatively strong, which can escalate the problem of numerical instabilities in large horizontal domains. In the present work, we propose an alternative inflow model for wind farm cluster simulations in RANS for neutral conditions near the ground but with a stable inversion layer, such a model reflects a conventional neutral ABL as commonly applied in large eddy simulations (Allaerts and Meyers, 2018; Kelly et al., 2019; Liu et al., 2021). The model does not use the global turbulence length scale limiter of Apsley and Castro (1997). Instead, the ABL height is set by a prescribed analytical temperature profile that includes an inversion height and strength. A temperature equation is not solved in order to maintain a steady-state inflow. Note that the use of an active temperature equation leads to an unsteady RANS model and we are interested in a steady-state RANS model.

The wind farm cluster validation case and the corresponding supervisory control and data acquisition (SCADA) measurements are discussed in Sect. 2. The numerical setup of the RANS simulations using ADs, the proposed AWF model and the new RANS inflow model are discussed in Sects. 3.1.1, 3.1.2 and 3.1.3, respectively. The numerical setup of the WRF simulations and engineering wake model calculations are presented in Sects. 3.2 and 3.3, respectively. The AWF model is verified and compared with the AD model in Sect. 4.1. In Sect. 4.2, the AWF model is compared with outputs from real-time simulations

using the WRF model (and a wind farm parametrization) and the TurbOPark engineering wake model (Nygaard et al., 2020;
Pedersen et al., 2022), and validated with SCADA measurements of a wind farm cluster consisting of three wind farms; two
are operated by Equinor, the other one by Ørsted.

## 2 Wind farm cluster validation case

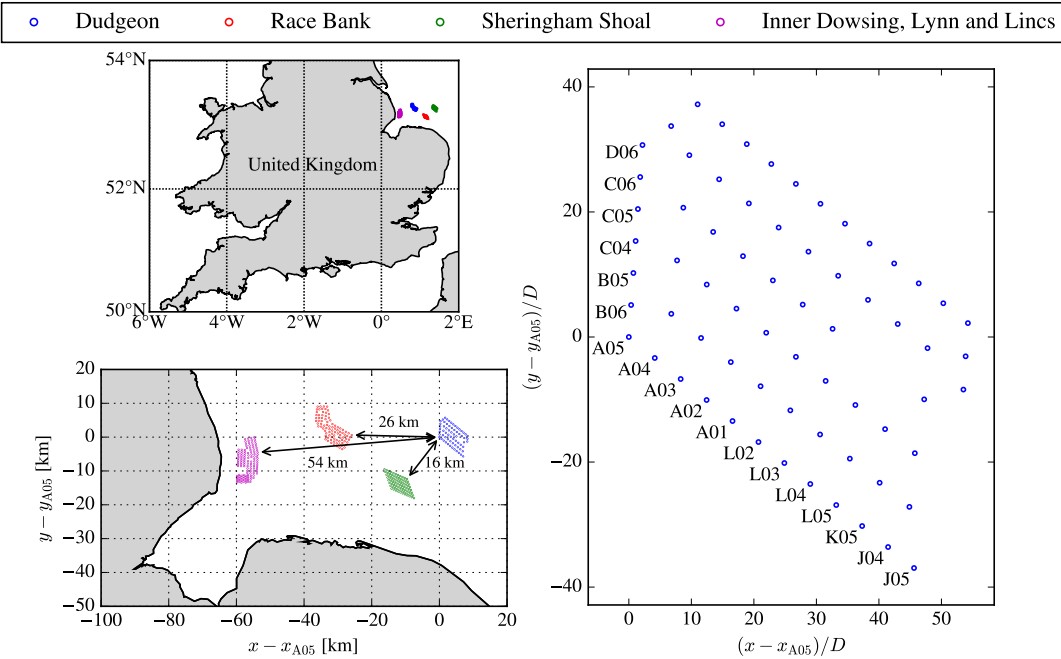

**Figure 1.** Wind farm cluster site.

Figure 1 depicts the investigated offshore wind farm cluster, which is located in the North Sea near the East coast of the
United Kingdom. There are four wind farms located in an area of about $70 \times 30$ km$^2$. SCADA measurements are made available
for the most Eastern wind farm, Dudgeon, and wind turbine power data from the front row wind turbines are employed to probe
the wind farm wakes of Sheringham Shoal and Race Bank, for wind directions around 235 and 270°, at wind farm interspacings
of 16 and 26 km, respectively. The most Western wind farm, located 54 km away from Dudgeon, consists of several smaller
wind farms (Inner Dowsing, Lynn and Lincs), which are not included in the present study. More details on the investigated
wind farms are listed in Table 1, where $A_{wf}$ is wind farm area, computed by a concave hull method, $N_{wt}$ is the number of wind
turbines, and $D$ and $z_H$ are wind turbine rotor diameter and hub height, respectively. The mean wind turbine interspacing is
calculated following Sørensen and Larsen (2021): $\sqrt{A_{wt}}/(D[\sqrt{N_{wt}}-1])$. The wind turbines are propriety to Siemens Gamesa
Renewable Energy (SGRE) and details on the thrust coefficient and power curves cannot be published. In the present work,

we investigate a below-rated inflow wind speed and the thrust curves of both wind turbines have typical below-rated thrust coefficient values.

| Wind farm | Operator | Layout type | $A_{\mathrm{wf}}$ [km$^2$] | Min/Mean spacing [$D$] | $N_{\mathrm{wt}}$ | Turbine | $D$ | $z_H$ |
|---|---|---|---|---|---|---|---|---|
| Dudgeon | Equinor | irregular | 49.9 | 5.1/6.4 | 67 | SGRE 6 MW | 154 | 102 |
| Race Bank | Ørsted | irregular | 57.1 | 4.8/5.7 | 91 | SGRE 6 MW | 154 | 102 |
| Sheringham Shoal | Equinor | regular | 32.3 | 6.1/6.3 | 88 | SGRE 3.6 MW | 107 | 80 |

**Table 1.** Wind farm cluster definition.

## 2.1 Measurements

This study uses three years of SCADA data from Dudgeon, from 2018-01-01 to 2021-01-01. For each turbine, the data are first averaged to ten minute periods and then, each period is kept only if it does not contain any: missing data, non-production or curtailment periods, low power values (< 100 kW) or low wind speed values (< 3 ms$^{-1}$). Following the filtering for each turbine, the ten minute period is kept if at least 64 out of 67 turbines remain, leading to a final availability of 56% of the data for the whole period.

In parallel, ERA5 data interpolated from the nearest ERA5 grid points to the wind farm location, for the same three-year period, is used to estimate the Obukhov length, $L$, from the wind speed at a height of 10 m, the temperature at a height of 2 m, and the sea surface temperature, applying an iterative method following Ott and Nielsen (2014). The data is divided into three atmospheric stability classes, namely unstable ($-200$ m $< L < 0$ m), neutral ($|L| >= 200$ m), and stable (0 m $<= L < 200$ m). This classification is then applied to the production data to get three data sets. The data of each stability class is then binned by 1 m s$^{-1}$ and 5° based on a reference wind speed and direction calculated by averaging the front row wind turbines for a specific sector. The latter is performed due to a lack of concurrent inflow measurements. The fact that we use the same wind turbine row to define the freestream and to probe the wake of an upstream wind farm means that is not possible to use the data set to validate the simulations results for the magnitude of wind farm wake losses. Therefore, the validation in Sect. 4.2 is only performed for the wind farm wake shape where the wind speed of the front row wind turbines are normalized with the wind speed averaged over the same turbines.

## 3 Simulation methodology

### 3.1 RANS

The wind farm simulations are performed with PyWakeEllipSys v3.0 (DTU Wind Energy, 2022b), which is a Python wrapper of the in-house flow solver EllipSys3D. EllipSys3D is a finite volume CFD code that has initially been developed by Michelsen (1992) and Sørensen (1994). The RANS model of EllipSys3D is used to simulate the steady-state wind farm flow under neutral atmospheric conditions including an ABL height, Coriolis forces, and a rough wall boundary with uniform roughness,

representing homogeneous terrain. The wind turbine forces are represented by two different methods. The baseline approach is an AD model (Réthoré et al., 2014) and is used to verify the proposed AWF method. Each method is discussed separately in Sects. 3.1.1 and 3.1.2, respectively.

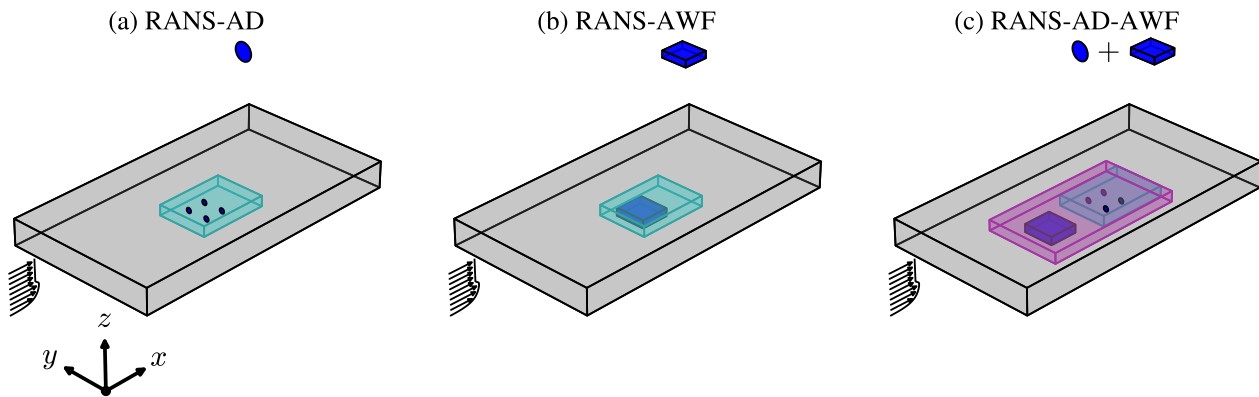

**Figure 2.** Sketch of different RANS domain types. Cyan and magenta boxes contain refined cells to resolve the wind turbine and farm wakes. Blue disks and boxes represent AD and AWF models, respectively.

In this work, different RANS flow domains are used to model wind farm clusters containing single, double and triple wind farms. A sketch of the flow domain types of the different applied actuator methods is depicted in Fig. 2. All flow domains are Cartesian grids where the inflow direction at the reference height is aligned with the $x$-axis and different wind directions are modeled by rotating the wind farm cluster, while maintaining the grid and the global inflow direction. The RANS-AD method (Fig. 2a) represents all wind turbines in a wind farm by ADs. Each wind turbine is treated as a single AD model, which has its own polar grid that is connected to the flow domain, and is also controlled independently. The cells around the ADs, as marked by the cyan box in Fig. 2a, are uniformly spaced in the horizontal plane using a fine resolution in order to resolve the wind turbine wakes. The RANS-AWF method (Fig. 2b follows a similar flow domain as the RANS-AD method; however, each wind farm is considered as a single AWF model that uses its own force controller. The wind turbine forces are distributed in a Cartesian grid that encapsulates the entire wind farm and this Cartesian grid is then connected to the flow domain grid. The refined area in the RANS-AWF method can be an order of magnitude coarser compared to the refined area for the RANS-AD model, which is investigated in detail in Sect. 4.1. The third domain type is depicted in Fig. 2c and represents the RANS-AD-AWF method, where both AD and AWF models are present. Since the AWF model does not require the fine spacing of the AD models, a second region in the flow domain is defined, as marked by the magenta box in Fig. 2c, where the cells are expanded in the horizontal direction while moving away from the cyan box, up to a maximum set spacing. It should be noted that the three methods as depicted in Fig. 2 can also be applied to simulate multiple wind farms. In this case, the size of the refined inner domain(s) are adjusted to resolve all wind turbine and farm wakes. An overview of the all the applied domains for the validation cases, as well as the computational effort is provided in Tab. 4.3.

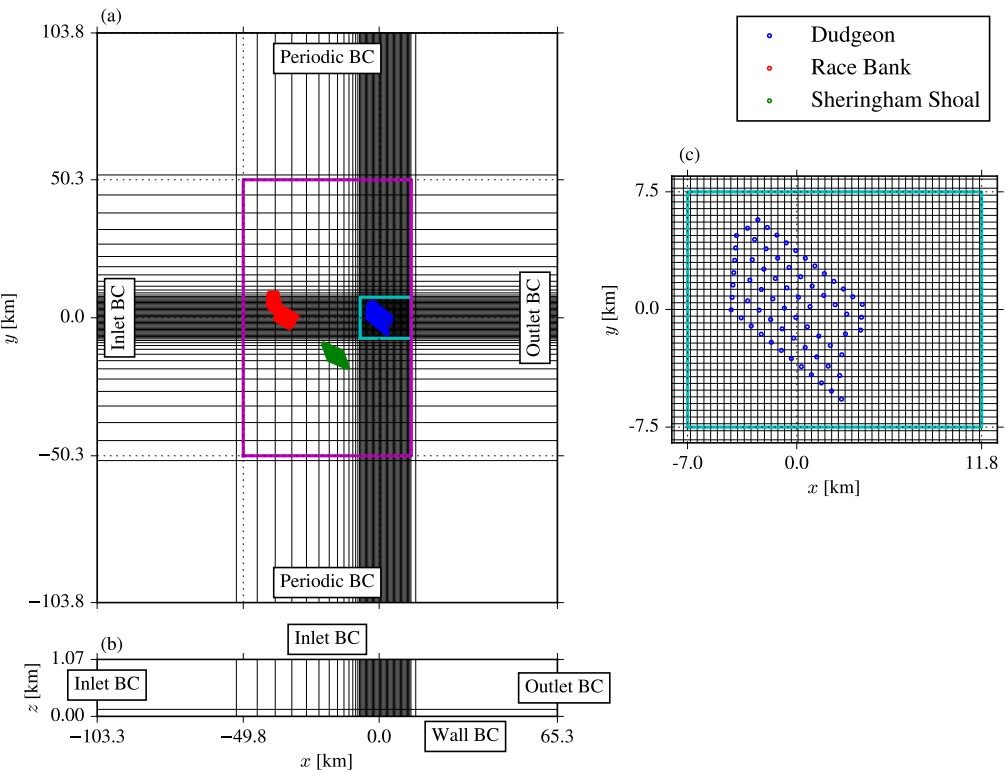

**Figure 3.** RANS grid and boundary conditions (BC) for the wind farm cluster validation case. Horizontal **(a)**, vertical **(b)** and a zoomed view around Dudgeon **(c)** are shown for every 32nd cell.

Figure 3 depicts a detailed description of the wind farm cluster validation case, where Dudgeon is modeled by ADs and the
other two wind farms, Sheringham Shoal and Race Bank, are modeled as AWFs (the RANS-AD-AWF flow domain type of
Fig.2c). The domain includes two areas with a refined horizontal spacing as marked by the cyan and magenta rectangles, as
shown in Fig. 3a. The cyan rectangle represents a uniform horizontal spacing of $D_{\mathrm{ref}}/8$ around the Dudgeon wind farm (van der
Laan et al., 2015c) and includes a sufficiently large area to resolve the wind farm flow for the wind directions of interests,
namely, 185–200°. Here, $D_{\mathrm{ref}}$ is the reference rotor diameter used to scale the grid and it is set to the smallest rotor diameter of
all the wind turbines (107 m). The magenta rectangle includes cells that are stretched towards the boundary and the maximum
horizontal cell size is limited to $2D_{\mathrm{ref}}$, which is the same as the horizontal resolution of the AWF model. The size of this area
is set to cover both Sheringham Shoal and Race Bank for the wind directions of interest. Outside the magenta rectangle, the
cells are further stretched with a maximum expansion ratio of 1.2 covering a distance of $500D_{\mathrm{ref}}$ (53.5 km). The total number
of cells in the $x$ and $y$ directions are 1920 and 2048, respectively. In the vertical, as depicted in Fig. 3b, the cells are stretched
while moving away from the ground and the first cell height is set to $D_{\mathrm{ref}}/200$ (0.54 m). The maximum cell size in the first
layer (up to $z = 3D_{\mathrm{ref}}$) is set to $D_{\mathrm{ref}}/6$, however, the cells in the rotor area and below are smaller than $D_{\mathrm{ref}}/8$. The height of

the domain is set to $10D_{\mathrm{ref}}$ and a total number of 64 cells are used in the vertical direction. This results in 252 million cells, which we divide into 960 blocks using a block edge size of 64 cells. We would have preferred to set a taller domain to avoid numerical blockage, however, there is a convergence issue when a tall domain height is applied together with an ABL inflow model that includes a low mixing region above the ABL height, which is further discussed in Sect. 3.1.3.

Figure 3a-b also depicts the boundary conditions, where inlet conditions are enforced over the inflow and upper domain faces; consequently the flow is lid-driven. Periodic boundary conditions are used at the lateral boundaries to be able to include wind veer. An outlet boundary condition is used at the outflow boundary, at which all gradients in the normal direction are assumed to be zero. The ground is modeled by a rough wall boundary condition following Sørensen et al. (2007).

### 3.1.1 AD model

The RANS-AD wind farm simulation setup is similar to the one used in previous work (van der Laan et al., 2015b). The wind turbines are modeled as ADs and each AD is represented by a polar grid using $10 \times 32$ cells in the radial and azimuthal directions, respectively. Each cell of the polar grid is connected to a set flow domain cells as discussed in detail by Réthoré et al. (2014). For each iteration, the thrust and tangential forces are calculated on the polar grid and are then injected in the flow domain grid as momentum sinks. The local velocities from the flow domain grid are returned to the polar grid to recalculate the AD forces and evaluate the wind turbine power. The magnitude of the thrust and tangential forces are controlled by a look-up table of alternative thrust and power coefficients that are based on a disk-averaged velocity to avoid the need for a freestream wind speed that is typically not known for interacting wind turbines; more details can be found in previous work (van der Laan et al., 2015a, 2019). For the model verification (Sect. 4.1), a general wind turbine model (van der Laan et al., 2022) is used with the same rotor diameter and hub height as the wind turbine from Dudgeon (SGRE 6 MW) but with a typical below rated thrust coefficient of 0.8, a power coefficient of 0.45 and a tip speed ratio of 8, which are all kept constant up to a rated wind speed of 10 m s$^{-1}$. The AD force distribution is modeled by the analytical AD force model from Sørensen et al. (2020). For the model validation (Sect. 4.2), we use an AD with a prescribed normalized thrust force distribution that is rescaled to obtain the desired thrust force magnitude (van der Laan et al., 2015a). For the SGRE 3.6 and 6 MW turbines, we use the thrust force distributions of the NREL-5MW (Jonkman et al., 2009) and DTU-10MW (Bak et al., 2013) reference wind turbines, respectively, calculated with previously performed detached eddy simulations (Réthoré et al., 2014; Bak et al., 2013), for a below rated wind speed of 8 m s$^{-1}$. This is because we lack information on the rotational speed of the SGRE wind turbines, which is a required input for the analytical rotor model of Sørensen et al. (2020) and we are lacking the corresponding airfoil data that is needed for a higher fidelity AD model based on a blade element momentum method. However, we do not expect a large influence on the wind turbine/farm wakes when using the AD model based on normalized thrust force distribution, as opposed to using AD models based on the analytical rotor model or airfoil data, because the prescribed normalized thrust force distribution takes into account a realistic radial force distribution and it employs the provided thrust coefficient curves of the SGRE wind turbines.

### 3.1.2 AWF model

The AWF model represents each wind farm as a single entity and its effect on the flow is modeled by a distributed thrust force:

$$F_{\mathrm{wf}}(x,y,z) = \frac{1}{2}\rho C_{T,\mathrm{wf}} A(x,y,z)|U|U_i, \tag{1}$$

where $\rho$ is the air density, $C_{T,\mathrm{wf}}$ is the wind farm thrust coefficient, $A(x,y,z)$ is the wind farm force distribution, representative of the wind turbine density and $U_i$ is the local velocity vector. Equation (1) is the same as that used to model a forest canopy drag force using $C_d = C_{T,\mathrm{wf}}/2$. In fact we employ the forest canopy model from EllipSys3D as used in previous work (Boudreault, 2015; Dellwik et al., 2019) but here we employ the same turbulence model as in the RANS-AD setup as opposed to the turbulence model developed for forest canopies (Sogachev et al., 2012; Boudreault, 2015; Dellwik et al., 2019).

One could employ additional terms in the turbulence model equations to account for the effect of under-resolving the wind farm layout in the AWF model when using large horizontal cells (Abkar and Porté-Agel, 2015). However, the literature is divided about whether this extra term should be zero (Volker et al., 2015), act as a source (Fitch et al., 2012; Abkar and Porté-Agel, 2015) or sink of turbulence kinetic energy (Sogachev et al., 2012). Investigating this is not in the scope of this study and so we do not use a source term of turbulence, partly because we already get reasonable results with only a momentum sink (Eq. 1).

**AWF model force distribution $A(x,y,z)$**

The drag force of the AWF model, $A(x,y,z)$, is distributed over a Cartesian grid encapsulating the wind farm. The vertical drag force distribution represents the rotor area slices following Abkar and Porté-Agel (2015). The horizontal drag force distribution represents the wind turbine density. Currently, when running the wind farm parametrization in the WRF model, the user sets the number of turbines per grid cell (for idealized cases) or inputs the turbine locations (real-time forcing cases) and so the number of turbines within the same grid cell are counted. Such counting can lead to grid-dependent results, particularly for large horizontal grid spacing and artificial velocity deficit shapes due to aliasing effects of the wind farm layout, as the WRF model is typically run at a horizontal grid spacing larger than 1 km. An example of this issue is illustrated in Sect. 4.1. To overcome this issue, we propose an alternative method to determine the horizontal drag force distribution of the wind farm by representing each wind turbine position as two-dimensional Gaussian functions, $f(x,y)$, instead of points:

$$f(x,y) = \exp\left(-\frac{[x-x_{\mathrm{wt}}]^2}{2\sigma_x^2} - \frac{[y-y_{\mathrm{wt}}]^2}{2\sigma_y^2}\right), \tag{2}$$

with $x_{\mathrm{wt}}$ and $y_{\mathrm{wt}}$ as the wind turbine positions and $\sigma_x$ and $\sigma_y$ as the standard deviations in the streamwise, $x$, and lateral, $y$, directions, respectively. We use $\sigma_x = 2\Delta$ and $\sigma_y = \max(2\Delta, D/4)$, with $\Delta$ as the horizontal grid spacing in the AWF model. The AWF horizontal grid spacing, $\Delta$, does not have to be the same as the horizontal spacing in the CFD grid, $\delta$. The use of $2\Delta$ is motivated in Sect. 4.1 and the max limiter is used in order to approach an AD model and avoid a too concentrated force; the latter could otherwise cause numerical problems. The AWF horizontal force distribution for each cell in the AWF model is obtained by superposing Eq. (2) for all wind turbines. This means that for a particular cell in the AWF grid, $(i,j)$, the wind

turbines outside this cell can also contribute to the drag force and cells that do not include a turbine can have non-zero force. In general, a smoothed wind farm layout is obtained as a smoothed wind farm force distribution and the individual wind turbines positions are resolved when a fine enough horizontal spacing is set. Another property of the Gaussian method is that a regular wind farm layout will always have a uniform thrust force distribution in the middle of the farm, which is not guaranteed when binning the number of wind turbines per cell. The magnitude of the drag distribution, $A(x, y, z)$, is not relevant because we

calibrate the wind farm thrust coefficient to get the desired total wind farm thrust force as discussed in the following section. The vertically integrated AWF horizontal force distribution of Race Bank and Sheringham Shoal are depicted in Fig. 4 for two different horizontal resolutions; $\Delta = D$ and $\Delta = 2D$. Figure 4 shows how the regular wind farm layout of Sheringham Shoal results in a uniformly distributed drag force when $\Delta$ is coarsened from $D$ to $2D$ by comparing Figs. 4b and 4d. In addition, values below 0.01 of the maximum force distribution are set to zero for numerical efficiency. The distributed drag forces are

stored in a separate Cartesian grid for each AWF model. The drag force of a CFD cell that lies within an AWF model is obtained by trilinear interpolation.

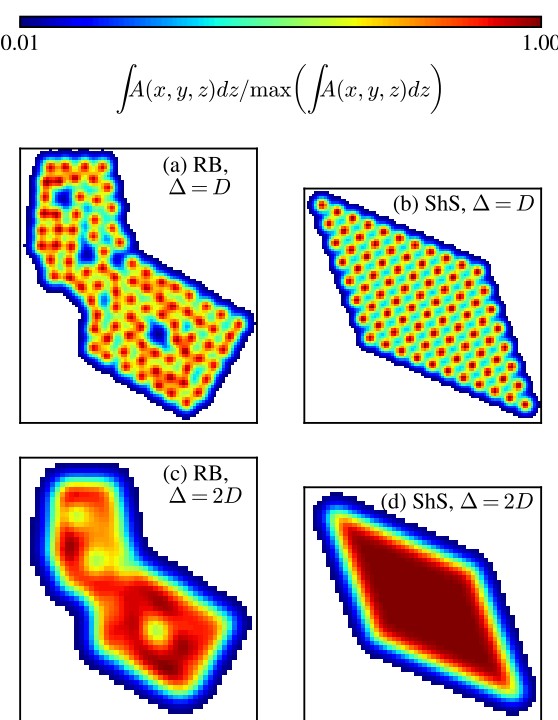

**Figure 4.** AWF horizontal drag force distribution integrated over the height for Race Bank (RB) **(a, c)** and Sheringham Shoal (ShS) **(b, d)** using different horizontal resolutions.

**AWF model force magnitude $C_{T,\mathrm{wf}}$**

Equation (1) depends on the local velocity and a local wind farm thrust coefficient, while a single wind turbine modeled as an AD with fixed forces can be set by a known thrust coefficient based on the freestream wind speed. Dellwik et al. (2019) calibrated the drag coefficient of a single tree numerically using a measured drag force as input. A related method was introduced by Abkar and Porté-Agel (2015), where the thrust force was multiplied by a correction parameter obtained by large-eddy simulations to correct the cell wind speed in a wind farm parametrization model to account for different wind farm layouts. For the AWF model, we follow a similar approach as Dellwik et al. (2019), where $C_{T,\mathrm{wf}}$ is calibrated using simulations of a single AWF model in order to obtain the wind farm thrust force determined by precursor RANS-AD simulations of the same wind farm. The wind farm thrust force depends on both the inflow wind speed through the wind turbine thrust coefficient curves and the inflow wind direction because of wind turbine interaction. Hence, $C_{T,\mathrm{wf}}$ is a function of both inflow wind speed and wind direction. For wind farm cluster simulations, the freestream wind direction and wind speed are undefined. Therefore, we employ a force controller for each AWF model, where $C_{T,\mathrm{wf}}$ is updated every solver iteration based on an interpolation of a look-up table where $C_{T,\mathrm{wf}}$ is a function of the AWF volume averaged wind speed and wind direction obtained from the AWF volume averaged velocity vector, $U_{\mathrm{AWF},i}$:

$$U_{\mathrm{AWF},i} = \frac{\int A U_i dV}{\int A dV}. \tag{3}$$

$U_{\mathrm{AWF},i}$ is weighted by the drag force distribution, $A$, so it does not include the buffer zone around the forest canopy and accounts only for the volume around the wind turbine position for fine spacing AWF models. Without the drag force distribution weighting, $U_{\mathrm{AWF},i}$ is typically over predicted. A similar approach was followed by Churchfield et al. (2017) for the application of the effective wind speed calculation for an actuator line model. The force control look-up table is created from the single AWF model calibration simulations used to determine $C_{T,\mathrm{wf}}$ from a desired total wind farm thrust force. The simulation steps necessary to perform the wind farm cluster validation case simulations are summarized below.

1. For each wind farm modeled as an AWF:

    (a) RANS-AD simulations are performed to calculate the total wind farm thrust force for a range of wind speed and directions ($U = 7, 8, 9$ m s$^{-1}$, $\phi = 175 - 310°$ with a 5° interval).

    (b) RANS-AWF simulations are performed using the same flow cases as step (a) in order to calculate $C_{T,\mathrm{wf}}$ and $C_{P,\mathrm{wf}}$ as a function of the AWF volume averaged wind speed and wind direction.

2. RANS-AD-AWF or RANS-AWF cluster simulations.

In steps 1 and 2, we neglect the influence of inhomogeneous inflow conditions on the wind farm thrust and power, as for example partial wind farm wake effects of the neighboring wind farms. However, the AWF model (as applied in Step 2 with a force controller) can partly respond to inhomogeneous conditions because the local thrust forces dependent on the local velocity, although variations in wind turbine thrust coefficients are not captured due to the use of a global wind farm thrust

coefficient. The impact of this assumption is investigated in Sect. 4.2.1. Step 1a is computationally expensive, especially when many different wind farms in a wind farm cluster are modeled as AWFs and each of them requires input from precursor RANS-AD wind farm simulations. The computational costs of step 1a could be alleviated by running RANS-AD wind farm simulations more efficiently (van der Laan et al., 2022). For example, one could run consecutive wind direction cases (applicable to homogeneous terrain) and consecutive wind speed cases (applicable to inflow models that include Reynolds-number similarity) both to save solver iterations, and employ wind farm layout symmetry to reduce the number of wind direction cases. In addition, one could also employ a fast engineering wake model to calculate the total wind farm thrust force, although this would most likely introduce a higher model uncertainty. The AWF model input can also be calculated with a RANS-based surrogate wind farm model, which is currently investigated in a follow up work (van der Laan et al., 2023).

### 3.1.3 Atmospheric inflow and turbulence models

We employ a two equation turbulence model in the form of a $k$-$\varepsilon$ model (Launder and Spalding, 1974):

$$\nu_T = C_\mu f_P \frac{k^2}{\varepsilon}, \tag{4}$$

$$\frac{Dk}{Dt} = \frac{\partial}{\partial x_j}\left[\left(\nu + \frac{\nu_T}{\sigma_k}\right)\frac{\partial k}{\partial x_j}\right] + \mathcal{P} - \varepsilon + \mathcal{B} + S_{k,\mathrm{amb}}, \tag{5}$$

$$\frac{D\varepsilon}{Dt} = \frac{\partial}{\partial x_j}\left[\left(\nu + \frac{\nu_T}{\sigma_\varepsilon}\right)\frac{\partial \varepsilon}{\partial x_j}\right] + (C_{\varepsilon,1}\mathcal{P} - C_{\varepsilon,2}\varepsilon + C_{\varepsilon,3}\mathcal{B})\frac{\varepsilon}{k} + S_{\varepsilon,\mathrm{amb}}, \tag{6}$$

with $x_j$ as the Cartesian coordinates, $\nu = 1.78406 \times 10^{-5}$ m$^2$s$^{-1}$ as the molecular viscosity of air, $\nu_T$ as the turbulent eddy viscosity, $k$ as the turbulent kinetic energy, and $\varepsilon$ as the dissipation of $k$. In addition, $\mathcal{P}$ is the mechanical production of turbulence, $\mathcal{B}$ buoyant turbulence production or destruction, and $S_{k,\mathrm{amb}}$ and $S_{\varepsilon,\mathrm{amb}}$ are ambient turbulence sources. The remaining unknowns in Eqs. (4)-(6) (except $f_P$) are turbulence model coefficients, here set as constants $(C_\mu, \sigma_k, \sigma_\varepsilon, C_{\varepsilon,2}) = (0.03, 1.0 1.3, 1.92)$, $C_{\varepsilon,1} = C_{\varepsilon,2} - \kappa^2/(\sigma_\varepsilon\sqrt{C_\mu})$ is used to enforce a balance with a logarithmic wind profile with $\kappa = 0.4$ as the von Kármán constant, and $C_{\varepsilon,3}$ is discussed later. In addition, the well known Boussinesq (1897) hypothesis is applied to define the relationship between the Reynolds-stresses and the strain-rate tensor. In absence of the buoyancy and without ambient sources, the model represents the $k$-$\varepsilon$-$f_P$ turbulence model, which has been developed for wind turbine wake simulation subjected to a neutral atmospheric surface layer (van der Laan et al., 2015c). The model includes a scalar function, $f_P$, which acts as a local turbulence length scale limiter in regions with high velocity gradients; i.e. the near wake. In previous works (van der Laan and Sørensen, 2017b; van der Laan et al., 2017), the $k$-$\varepsilon$-$f_P$ model was combined with an atmospheric boundary layer inflow model employing a constant pressure gradient, Coriolis forces and the global turbulence length scale limiter of Apsley and Castro (1997) in order to simulate the effect of an idealized ABL on a wind farm. In addition, ambient turbulence sources were used to avoid zero turbulence quantities above the ABL. This ABL model does not require a temperature equation nor a buoyancy source term and can represent stable conditions by setting a low value of the maximum turbulence length scale, $\ell_{\max}$, by replacing $C_{\varepsilon,1}$ with $C_{\varepsilon,1}^* = C_{\varepsilon,1} + (C_{\varepsilon,2} - C_{\varepsilon,1})\ell/\ell_{\max}$, as introduced by Apsley and Castro (1997). Note that the turbulence length scale is a model parameter defined as $\ell \equiv C_\mu^{3/4} k^{3/2}/\varepsilon$. When the global turbulence length scale limiter of Apsley and Castro (1997) is applied to three-dimensional flows containing turbulence length scales larger than the maximum

values set by $\ell_{max}$, one can observe non-physical behavior in the solution and also numerical problems (van der Laan et al., 2017). This can occur for flow over complex terrain with large hills, for large wind farms and clusters of wind farms, as investigated in the present work. The non-physical behavior typically leads to an artificially slow wake recovery. Therefore, we propose an alternative atmospheric inflow that sets the boundary layer height by an inversion layer using a prescribed potential temperature profile in combination with a non-zero $\mathcal{B}$ instead of using the global turbulence length scale limiter of Apsley and Castro (1997). The new model does not employ an active temperature equation in order to maintain a steady-state model because the use of an active temperature equation in combination with a non-linear temperature profile results in an unsteady inflow model, where the boundary layer height keeps growing with time.

We employ an analytically prescribed potential temperature profile, $\theta$, by defining a zero and a constant temperature gradient at the ground and above the inversion height, $z_i$, respectively, including a smooth transition in between the two regions:

$$\frac{\partial \theta}{\partial z} = \frac{1}{2} \left[ 1 + \tanh \left( \frac{z/z_i - 1}{z_T/z_i} \right) \right] \frac{\partial \theta}{\partial z} \bigg|_c , \tag{7}$$

where $\partial \theta / \partial z|_c$ is the constant temperature gradient above $z_i$ and $z_T$ is half the width of the transition zone around $z_i$. The latter is clear when taking a first order Taylor expansion of Eq. 7: $\partial \theta / \partial z \approx 1/2[1 + (z - z_i)/z_T] \partial \theta / \partial z|_c$ and setting $z = z_i \pm z_T$. A temperature profile can be obtained by integrating over the height $z$ and using $\theta(z = 0) = \theta_0$ as the surface temperature:

$$\frac{\theta}{\theta_0} = 1 + \frac{1}{2\theta_0} \frac{\partial \theta}{\partial z} \bigg|_c \left[ z + z_T \ln \left( \frac{\cosh \left( \frac{z - z_i}{z_T} \right)}{\cosh \left( z_i / z_T \right)} \right) \right] . \tag{8}$$

There is a numerical issue with $\cosh(z)$ going to infinity for large $z$, which can be addressed by rewriting $\cosh(z)$ as $(1 + e^{-2x})/(2e^{-x})$:

$$\theta = \theta_0 + \left( z - z_i + \frac{z_T}{2} \ln \left[ \frac{1 + e^{2(z_i - z)/z_T}}{1 + e^{-2z_i/z_T}} \right] \right) \frac{\partial \theta}{\partial z} \bigg|_c . \tag{9}$$

We set $z_T/z_i$ equal to 0.2; larger values would result in more smoothing (wider transition) and other values could be investigated in future work, though.

The effect of the prescribed temperature is represented by a buoyancy source term in the $k$ and $\varepsilon$ transport Eqs. (5)-(6):

$$\mathcal{B} = -\frac{\nu_T}{\sigma_\theta} \frac{g}{\theta} \frac{\partial \theta}{\partial z} , \tag{10}$$

with $\sigma_\theta = 0.74$ as the Prandtl number for temperature and $g = 9.81$ ms$^{-2}$ as the gravitational acceleration constant. We do not use a buoyancy source term in the momentum equation and a constant air density of 1.225 kg m$^{-3}$ is employed in the present work as we only simulate wind farms in flat terrain. The buoyancy source term in the $\varepsilon$ equation, (Eq. 6): is multiplied by a constant $C_{\varepsilon,3}$:

$$C_{\varepsilon,3} = 1 + C_{\varepsilon,1} - C_{\varepsilon,2}, \tag{11}$$

which is based on the transient ABL model of Sogachev et al. (2012) without the global turbulence length scale limiter $\ell_{max}$ (or using $\ell_{max} = \infty$). In addition to the buoyancy sources, we also add ambient sources terms (Spalart and Rumsey, 2007),

$S_{k,\mathrm{amb}}$ and $S_{\varepsilon,\mathrm{amb}}$ to the $k$ and $\varepsilon$ transport Eqs. (5)-(6), respectively, in order to prevent zero values of $k$ and $\varepsilon$ above the ABL,
similar to van der Laan et al. (2020), but replacing $\ell_{\max}$ with $z_i$:

$$S_{k,\mathrm{amb}} = \varepsilon_{\mathrm{amb}}, \qquad S_{\varepsilon,\mathrm{amb}} = C_{\varepsilon,2} \frac{\varepsilon_{\mathrm{amb}}^2}{k_{\mathrm{amb}}}, \tag{12}$$

$$\ell_{\mathrm{amb}} = C_{\mathrm{amb}} z_i \qquad k_{\mathrm{amb}} = \frac{3}{2} I_{\mathrm{amb}}^2 G^2, \qquad \varepsilon_{\mathrm{amb}} = C_\mu^{3/4} \frac{k_{\mathrm{amb}}^{\frac{3}{2}}}{\ell_{\mathrm{amb}}},$$

with $k_{\mathrm{amb}}$, $\varepsilon_{\mathrm{amb}}$, $\ell_{\mathrm{amb}}$ and $I_{\mathrm{amb}}$ as the ambient values for $k$, $\varepsilon$, the turbulence length scale and turbulence intensity above the
ABL, respectively. We set $C_{\mathrm{amb}} = 10^{-7}$ and $I_{\mathrm{amb}} = 10^{-5}$, which are lower values compared to previous work (van der Laan
et al., 2020), but are necessary in order to get numerically stable results and are still low enough to not have an influence the
inflow profiles.

Since we lack measurements of the freestream, we need an alternative source to determine the input parameters for the
inflow model. The New European Wind Atlas (NEWA) database (Dörenkämper et al., 2020) comprises 30 years (1989–2018)
of simulated output of the WRF model for Europe. We extended the simulated period to cover 2019 and 2020 using the same
model configuration of NEWA but only within NEWA's Great Britain subdomain. We extract data for a three year time period:
01-01-2018–01-01-2021 at the grid point closest to the wind farm cluster center at latitude and longitude of 53.21647 and
1.10947°. The grid point is only 1.12 km from the cluster center. The data is filtered for the same criteria as the measurements;
a wind speed bin of 7–9 m s⁻¹ (at $z = 100$ m) and a Obukhov length range of $|L| \geq 200$ m. Based on the WRF model output
for these three years, the boundary layer height is 512 m, the wall temperature (skin temperature) is 285 K and the turbulence
intensity based on $k$, $I_{\mathrm{ref}}$, at $z = 100$ is 4.4%. For the RANS ABL inflow model, we use the same wall temperature and $I_{\mathrm{ref}}$ at
$z = 102$ m, and an inversion height of $z_i = 1000$ m and a inversion strength of 0.005 K m⁻¹. This results in a wind speed ABL
height of about 700 m, as depicted in Fig. 5, where results of a precursor simulation are shown employing EllipSys1D (van der
Laan and Sørensen, 2017a). Fig. 5 also includes the selected NEWA data simulated by the WRF model (only data up to
500 m was extracted) and a neutral atmospheric surface layer (ASL) using the same $I_{\mathrm{ref}}$ and $U_{\mathrm{ref}}$. The result from the WRF
model and the prescribed temperature predict a similar wind speed, wind veer and turbulence intensity around rotor area,
but the inflow profiles are quite different above the wind turbine. The potential temperature profile from the WRF model
only matches near the ground and indicates stable conditions above 75 m or a very shallow inversion height. Setting a lower
$z_i$ in the prescribed temperature model in order to obtain an ABL height closer to that output by the WRF simulations is
possible. However, numerical convergence problems can occur when such a shallow ABL inflow is applied for wind farms
with large wind turbines, as discussed in more detail in Appendix B. The Coriolis parameter is set to $1.168 \times 10^{-4}$ s⁻¹ based
on the latitude of the wind farm cluster center. The geostrophic wind speed and roughness length are found by an optimization
procedure employing the previously mentioned parameters and using a reference wind speed, $U_{\mathrm{ref}}$ of 8 m s⁻¹ at $z = 102$ m.
The input and derived parameters are summarized in Tab. 2.

In the present work, we only use one inflow profile for all simulations for simplicity, and also when the inflow wind speed
cases are not equal to 8 m s⁻¹. For example, the ADs are controlled using a look-up table based on force calibration simula-
tions (van der Laan et al., 2015a), for which the entire wind speed range of the operational regime is computed for a single

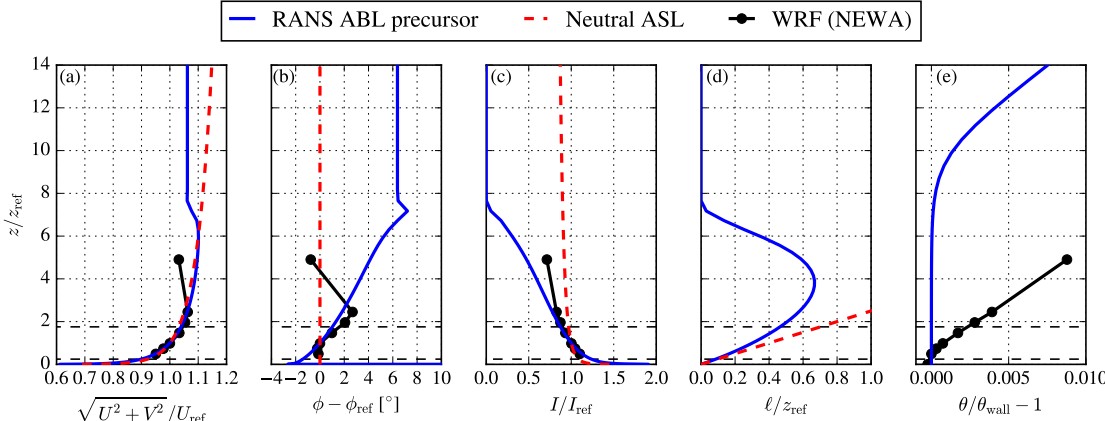

**Figure 5.** Atmospheric inflow model precursor results employing a prescribed temperature profile compared to a neutral ASL solution and results from the WRF model. Results are shown in terms of wind speed **(a)**, relative wind direction **(b)**, turbulence intensity **(c)**, turbulence length scale **(d)** and potential temperature **(e)**. Horizontal dashed black lines depict the swept rotor area of the 6 MW wind turbine.

| | Input parameters | | | | | | Derived parameters | |
|---|---|---|---|---|---|---|---|---|
| $I_{ref}$ | $U_{ref}$ [m s$^{-1}$] | $z_{ref}$ [m] | $f_c$ [s$^{-1}$] | $z_i$ [m] | $\theta_0$ [K] | $\partial\theta/\partial z\|_c$ [K m$^{-1}$] | $G$ [m s$^{-1}$] | $z_0$ [m] |
| 0.044 | 8 | 102 | $1.168 \times 10^{-4}$ | 1000 | 285 | $5 \times 10^{-3}$ | 8.50 | $3.25 \times 10^{-5}$ |

**Table 2.** Summary of input and derived parameters for the ABL inflow model.

wind turbine by using different $C_T$ values while keeping the inflow constant. This means that we assume Reynolds-number similarity, which is not valid for the prescribed temperature inflow model because it follows a Rossby- and Zilitinkevich number similarity, as shown in Appendix A. However, since the AD controller is only applied for wind farm flows around a wind

speed of 8 m s$^{-1}$, we neglect the effect of small variations in these dimensionless numbers.

The new inflow model still includes a low eddy viscosity region above the ABL height similar to the ABL model based on $\ell_{max}$, as shown by the small turbulence length scale in Fig. 5d for $z > 7z_{ref}$. If this region is included in the inflow for a three-dimensional simulation of a large wind farm cluster (e.g. in order of $200 \times 200$ km$^2$ or larger), then numerical instabilities can occur (van der Laan et al., 2017). Here, the largest horizontal domain is $169 \times 208$ km$^2$, which is just small enough to avoid

numerical instabilities if the domain height is set to $10D_{ref}$ (about 1 km). Furthermore, if the prescribed temperature model is employed with an inversion height that results in a shallow ABL, where the wind turbines are (partially) operating in the low eddy viscosity region then numerical problems can occur as well. Appendix B illustrates the issue of employing shallow ABL inflows to wind farm cluster simulations and the new prescribed temperature inflow model is also compared with the inflow model based on $\ell_{max}$. We find that the use of Rayleigh damping in a steady-state method as RANS, does not remove

the numerical instabilities, possibly because Rayleigh damping methods have been developed for transient simulations (Durran

and Klemp, 1983). The numerical issues associated with the low eddy viscosity region above the ABL need to be further investigated in future work.

## 3.2 WRF

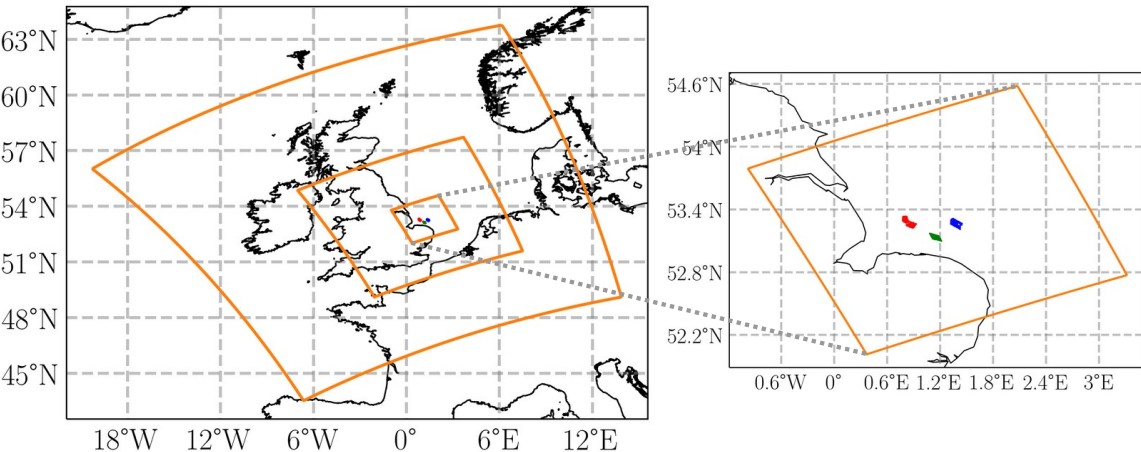

**Figure 6.** Depiction of the WRF model domains (orange frames): outermost parent domain (9 km grid spacing) with two nested inner domains (3 km and 1 km grid spacing, respectively). The location of the Dudgeon, Sheringham Shoal and Race Bank wind farms are indicated in blue, green and red, respectively.

In the previous section, output from a mesoscale long run using the WRF model was used to derive input parameters for the RANS inflow model. Another set of mesoscale simulations using the WRF model was performed on the Dudgeon wind farm area (including the Dudgeon, Sheringham Shoal and Race Bank wind farms) but using an implementation of the WRF model version 3.7.1, in which the explicit wake parameterization (EWP) is included (Volker et al., 2015). The EWP represents mesoscale wind farm effects on the atmospheric flow. Turbine induced drag forces are formulated as grid-cell averaged forces, while wind turbine induced turbulence is treated via an explicit sub-grid scale turbulence diffusion formulation (Volker et al., 2015). In contrast to WRF's native wind farm parameterization (Fitch et al., 2012), no explicit source of turbulent kinetic energy (TKE) is added to the TKE equation by the EWP. The simulation uses a one-way nested domain setup with three domains (Fig. 6). The innermost domain has a grid spacing of 1 km with 216 x 216 grid points. More details about model configurations as well as initial and boundary conditions are provided in Appendix D. This simulation is aimed to be compared with PyWakeEllipSys based modeling approaches of wind farm cluster wakes, which use idealized representations of the ABL and does not include mesoscale effects. To provide a fair comparison, the simulation period (calendar year of 2018) has been screened for weather episodes that fulfill the following conditions at the WRF grid-cell center closest to the L02 turbine of Dudgeon (Fig. 1): (1) inflow wind speed at hub-height of 7–9 m s$^{-1}$, (2) inflow direction of 232.5–237.5° and (3) near-neutral stability conditions (Obukhov Length $|L| > 200$ m). To avoid the inclusion of isolated cases, all conditions are required to

persist for at least one hour. This resulted in a sample size of 31 ten minute temporal snapshots (instantaneous wind speeds)
from three distinct weather events in 2018. Furthermore, a reference simulation without wind farms covering the same calendar
period is run and the same 31 ten minute temporal snapshots are extracted for normalization of the wind speed results from the
simulation with wind farms. This is performed to correct for the impact of the background mesoscale flow, e.g. coastal effects
or large-scale wind speed gradients in the horizontal wake profile.

### 3.3 TurbOPark engineering wake model

The TurbOPark model from Nygaard et al. (2020) and Pedersen et al. (2022) is an analytical wake model optimized for
simulating large offshore wind farms and farm-to-farm interaction. It employs a Gaussian wake deficit profile, but differs in its
definition of the wake expansion from other analytical models by using non-linear streamwise wake expansion in contrast to the
more commonly used linear expansion. The non-linearity derives from assuming that the wake expansion rate is proportional
to the local turbulence intensity, which is assumed to be a combination of atmospheric and turbine's added turbulence. In
combination with Frandsen's model for the streamwise attenuation of turbulence intensity (Frandsen, 2007), the wake expands
quickly near the rotor but progressively slower moving further downstream. This conserves wakes deficits over large distances
making the model suitable for assessing wake losses in large arrays or between farms.

The TurbOPark wake model is implemented in DTU Wind's open-source wind farm simulation tool PyWake (DTU Wind
Energy, 2022a). PyWake is fully modular, so gives complete freedom how wake, blockage and other sub-models are combined
to define the engineering wind farm model for the AEP simulation. The local averaged disc velocity is used to determine
thrust, power and deficits, whereas the freestream streamwise turbulence intensity of 5.5%, estimated from the turbulence
intensity based on the TKE as 4.4/0.8 (van der Laan et al., 2015c), is used at all turbines to determine wake expansion and
the superposition of wake added turbulence is not taken into account. A wind farm optimized version of the self-similar
single turbine blockage model (Troldborg and Meyer Forsting, 2017), derived from RANS simulations of multiple full-scale
turbines, accounts for wind farm blockage effects including speed-ups. A mirror plane models the ground effect for both
wakes and blockage. Wake deficits are superposed by the root-sum-square, whereas blockage effects are linearly summed, as
they constitute small perturbations. The total wind farm flow field is obtained by summing blockage and wake contributions.
The flow solution is obtained iteratively, as deficit information needs to be passed up- and downstream. The computation is
accelerated by initializing the solution with a wakes-only simulation. Around cut-in and cut-off wind speed, the discontinuous
nature of the thrust curve may spoil convergence, as certain turbines switch on and off between iterations. This is avoided by
identifying unstable turbines and switching them off permanently.

TurbOPark as implemented in PyWake is used with two different setups: one reflects the original model setup from Nygaard
et al. (2020) and Pedersen et al. (2022), and the other is a revised setup where the ground model for the wake deficit is
switched off and a wake expansion coefficient of 0.06 (instead of 0.04) is employed, as we find that the original model setup
overpredicts wind farm wake effects. The revised setup is one way of reducing these effects such that the results compare better
with the simulated results from RANS and the WRF model for the investigated double wind farm case, as shown in Sect. 4.2.1.

However, other setups, with linear wake summation for instance, could give as convincing predictions if tuned. An overview of the PyWake setups are given in Appendix C.

## 4 Results and Discussion

### 4.1 Model verification

#### 4.1.1 Horizontal drag distribution: Binning vs Gaussian methods

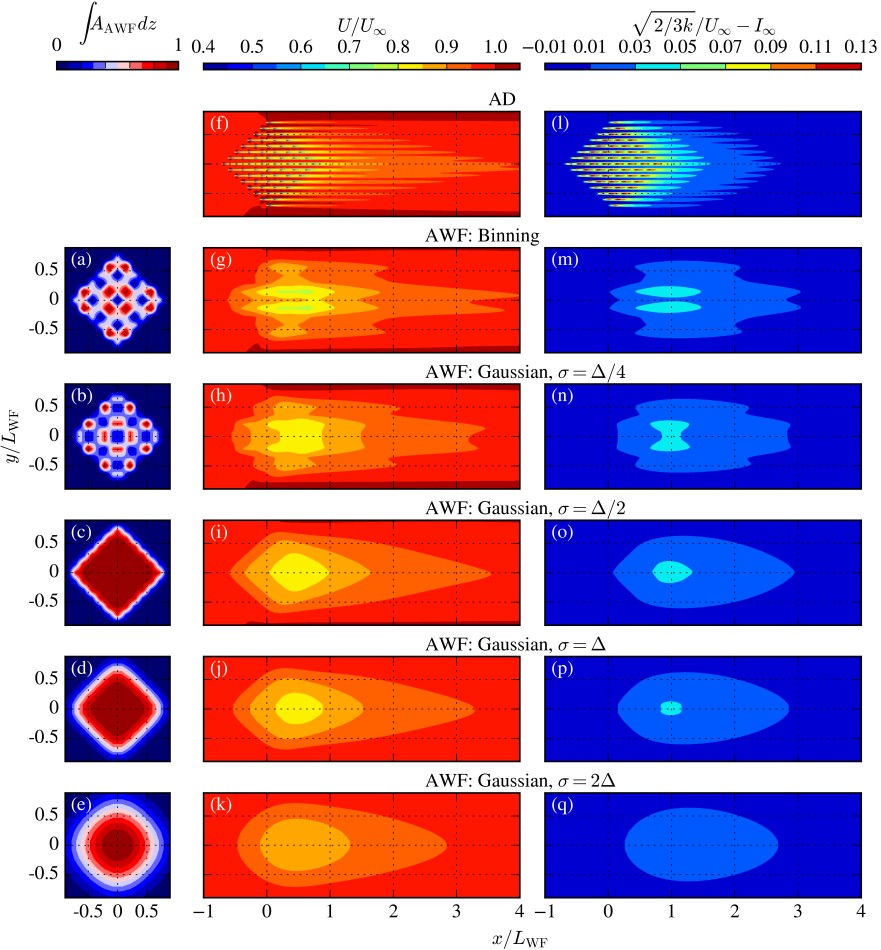

**Figure 7.** The effect of different horizontal drag force distribution methods for a coarse AWF model ($\Delta = 8D$): Binning vs Gaussian, for a rotated square wind farm. Contours of integrated canopy density **(a-d)**, streamwise velocity **(e-i)** and turbulence intensity **(j-n)** at hub height. AD results **(e, j)** are depicted as a reference.

The horizontal drag distribution in the AWF model represents the wind turbine density per cell. If large horizontal cells are used in the AWF model, e.g. in the order of the wind turbine interspacing in a wind farm, then the AWF grid orientation with respect to the wind farm layout can have a large influence on the wind farm wake shape when counting or binning the number wind turbines per cell. We refer to this method as the binning method. One can solve the issue by using a superposition of two dimensional Gaussian functions for the wind turbines to represent the number of wind turbines per cell. To illustrate the difference between the binning and the Gaussian methods, a square wind farm with $8 \times 8$ wind turbines using a spacing of $8D$ is simulated for a wind direction aligned with the diagonal. In practice, we simply rotate a square wind farm layout by $45°$. The wind farm is simulated using RANS-AD (as a reference) and five RANS-AWF models with a large horizontal grid spacing in the drag distribution grid ($\Delta = 8D$); one using the binning method and four using the Gaussian method with smoothing by employing different standard deviations: $\sigma = \sigma_x = \sigma_y = \{\Delta/4, \Delta/2, \Delta, 2\Delta\}$. The flow domain for the AWF models is the same as the one used for AD model; namely a horizontal spacing of $D/8$ at the wind farm location and a maximum of $1D$ in the wind farm wake up to four wind farm lengths, $L_{\text{WF}}$, downstream. Figure 7 shows results of these simulations in terms of streamwise velocity normalized by the freestream (Figs. 7f-k) and added wake turbulence intensity (Figs. 7l-q), both quantities are extracted at hub height. The vertically integrated drag distribution of the AWF models (after interpolation to the CFD grid) are shown in Figs. 7a-e. When the binning method is applied, the drag distribution becomes inhomogeneous due to aliasing effects, see Fig. 7a. This results in artificial wind farm wake shapes, as depicted in Figs. 7g and 7m compared to the AD simulation shown in Figs. 7f and 7l. The same problem occurs with the Gaussian method when not enough smoothing is applied, e.g., the case for $\sigma = \Delta/4$ (Figs. 7b,h,n). However, when more smoothing is used, i.e., $\sigma = \Delta/2$ and $\sigma = \Delta$, the horizontal drag force distribution becomes uniform, as shown in Figs. 7c-e (as expected for a regular layout), and the artificial wake shape disappears (Figs. 7i-k and 7o-q).

The results shown in Fig. 7 are also depicted in Fig. 8 but in terms of lateral profiles of streamwise velocity at hub height and at three different downstream locations. It is clear that by using the binning and the Gaussian method without enough smoothing, $\sigma = \Delta/4$, artificial wake shapes appear, which are not present in the simulations using ADs. The results using the largest smoothing, $\sigma = 2\Delta$, reduces the accuracy with respect to the AD simulation because the effective resolution also reduces with an increased level of smoothing. However, using $\sigma = \Delta$ can lead to a checkerboard horizontal drag force distribution for a particular setup, e.g., for a non-rotated square wind farm layout with $8 \times 8$ wind turbines, $8D$ spacing and $\Delta = 2D$. This problem disappears when using $\sigma = 2\Delta$. From a numerical point of view, smoothing body forces with a Gaussian in a numerical simulation should use a minimal value of $\sigma = 2\Delta$ in order to prevent wiggles in the solution. This is based on the Gaussian smoothing used to represent the blade forces in actuator line models (Troldborg, 2008; Forsting and Troldborg, 2020). Therefore, we apply a smoothing of $\sigma = 2\Delta$ for all other simulations using the AWF model. If one would like to increase the accuracy of the AWF model then it is recommended to refine $\Delta$ with $\sigma = 2\Delta$ and not solely reducing $\sigma$.

### 4.1.2 Horizontal drag distribution: AWF grid and CFD grid spacing

The effect of the horizontal resolution in the AWF grid, $\Delta$, is depicted in Fig. 9 using a wind farm with $8 \times 8$ wind turbines, a turbine spacing of $8D$ and a row aligned wind direction. Results of six AWF models with different $\Delta$ values are shown and

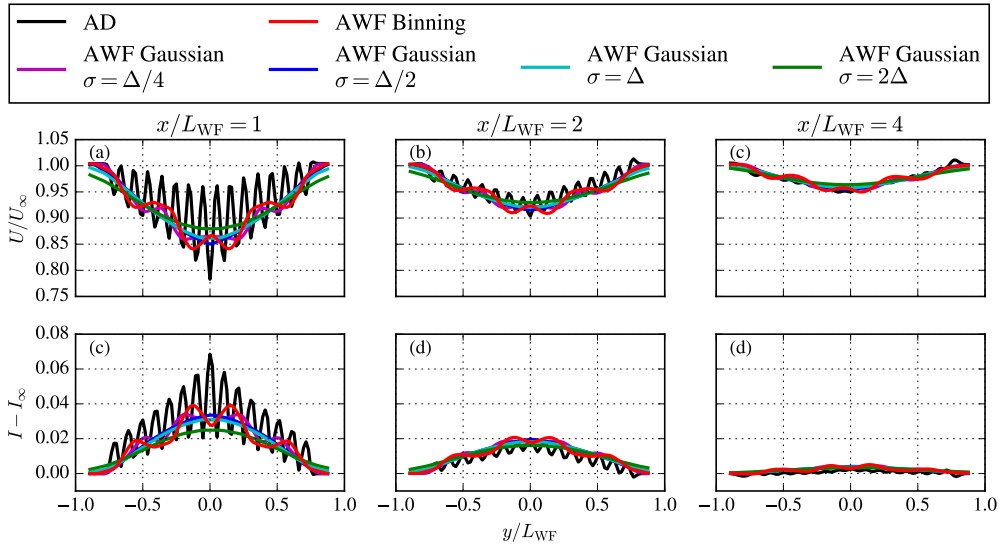

**Figure 8.** The effect of different horizontal drag force distribution methods for a coarse AWF model ($\Delta = 8D$): Binning vs Gaussian, for a rotated square wind farm. Lateral profiles of streamwise velocity **(a-c)** and turbulence intensity **(d-f)** at hub height, at three different downstream distances are shown. AD results are depicted as a reference.

the results are compared with results from the AD model (using the same CFD grid spacing as the AD model, $\delta = D/8$) in terms of streamwise velocity normalized by the freestream (Figs. 9g-m) and added wake turbulence intensity (Figs. 9n-t), both extracted at hub height. The vertically integrated drag distribution of the AWF models are shown in Figs. 9a-f. When refining the horizontal spacing in the AWF model, the solution approaches the one of the AD model. However, there are still differences

between the AWF model with $\Delta/8$ and the AD model mainly at the wind farm location, as shown in Figs. 9g-h and Figs. 9n-o. If a closer match is desired one would need to further refine $\Delta$. The need for a finer AWF grid spacing in order to approach an AD simulation results is not a surprise because the drag distribution in the AWF model is represented by a Cartesian grid and thus more cells than 8 are required to represent a rotor disk area by square cells. This is also why our AD model uses a polar grid for each wind turbine in the RANS-AD setup. When $\Delta \geq D$ or $\Delta \geq s/8$, with $s$ as the wind turbine interspacing,

the individual wind turbines are no longer visible in the horizontal drag distribution of the AWF model, as shown in Fig. 9.

The difference between the AD and the AWF simulation results are depicted in Fig. 10 in terms of horizontal velocity, Figs. 10a-b, and turbulence intensity, Figs. 10c-d. A range of wind directions are simulated between 270–315° with an interval of 5°. For each wind direction, the mean and maximum absolute difference between the AD and AWF models is calculated from cross planes centered at $(y, z) = (0, z_H)$ with a width $L_{\text{WF}}$ and a height $D$, for a range of downstream locations to

485 estimate the impact of $\Delta$ on a potential downstream wind farm. Subsequently, the results are either averaged over the wind directions, Fig. 10a, c, or filtered for the maximum absolute difference, Fig. 10b,d. In addition, results are shown using the same CFD grid as the AD model, $\delta = D/8$, and when the AWF model spacing is the same as the CFD grid spacing, $\Delta = \delta$. The first provides a more fair comparison while the latter is used in practice when applying the AWF model in a wind farm

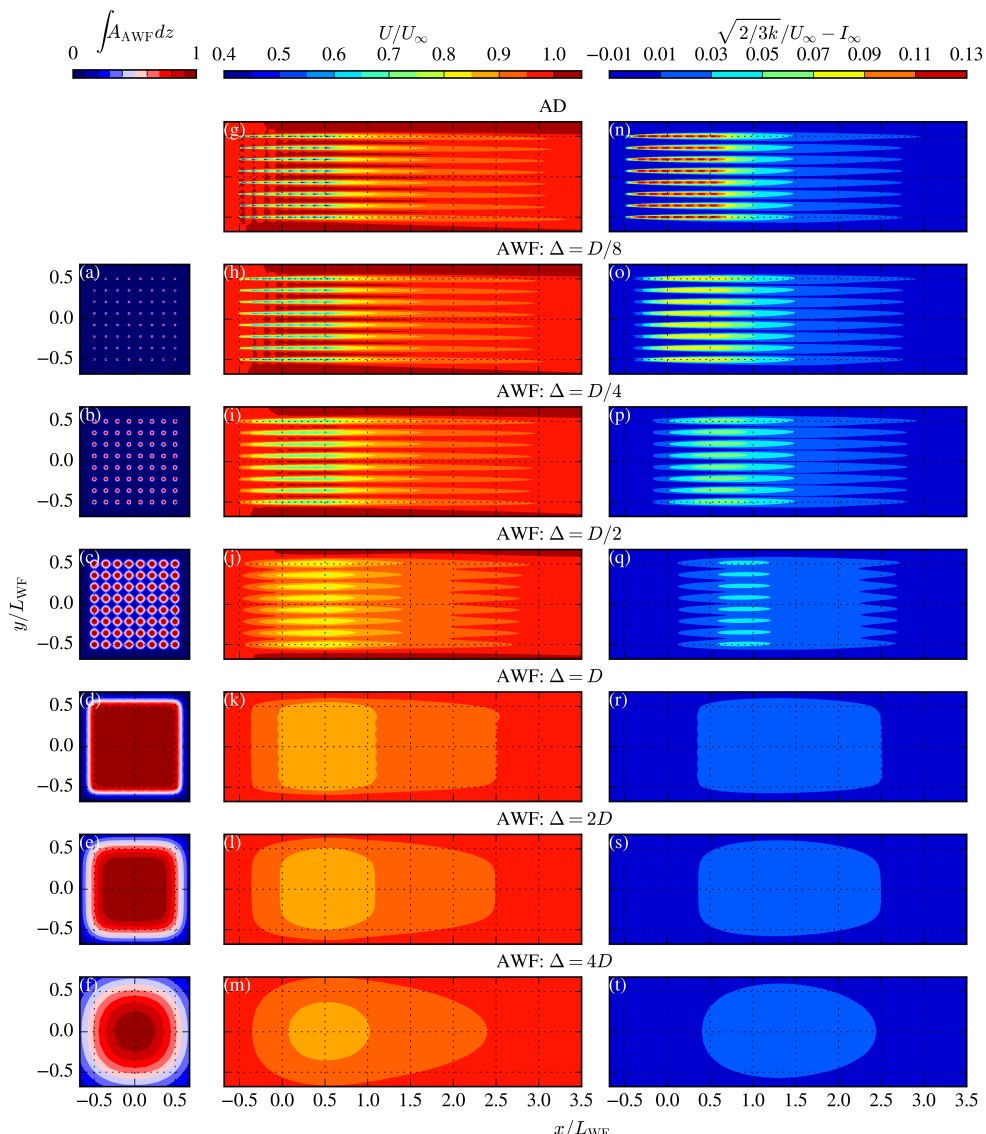

**Figure 9.** The effect of different horizontal spacing in the AWF model for a square wind farm. Contours of integrated canopy density **(a-f)**, streamwise velocity **(g-m)** and turbulence intensity **(n-t)** at hub height. AD results **(g, n)** are depicted as a reference.

cluster. As expected, the difference between the AWF and AD models increases with coarsening the horizontal AWF model resolution, $\Delta$. However, the difference is not much influenced by coarsening the CFD grid by using $\Delta = \delta$; the main effects are observed for the large AWF model spacing of $\Delta = 4D$ and $8D$. The errors are the largest at the wind farm area, which is not our area of interest when upstream wind farms are modeled as AWFs. One could select a value for $\Delta$ depending on the desired accuracy. For the present work, we select $\Delta = 2D$, which results in a mean error in streamwise velocity and turbulence intensity of $\pm 0.4\%$ and $0.3\%$, respectively, for the wind farm wake at $x > 2L_{\mathrm{WF}}$. The maximum absolute errors at $x > 2L_{\mathrm{WF}}$

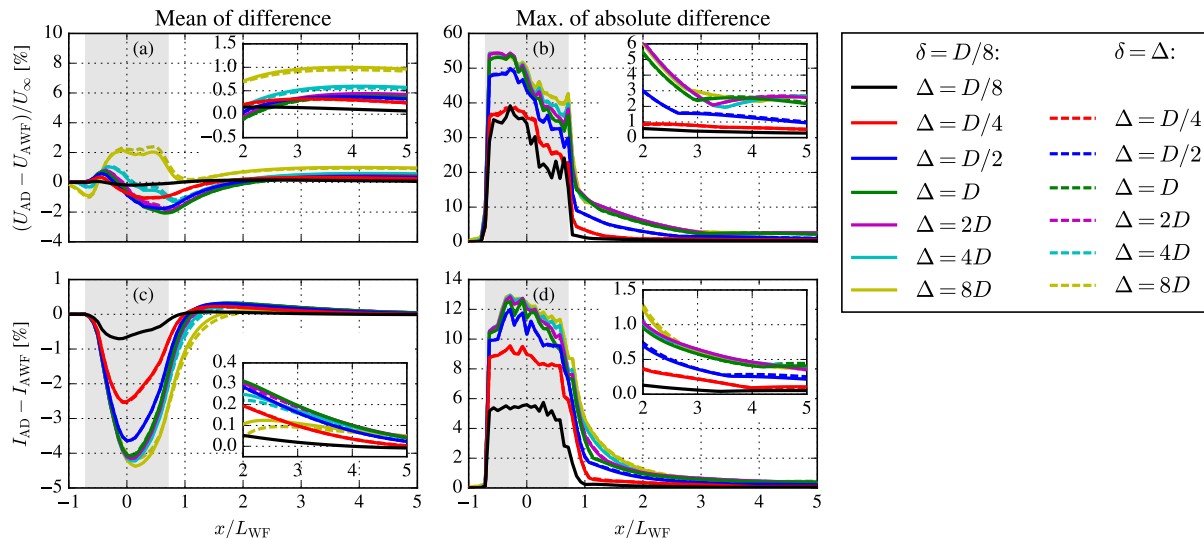

**Figure 10.** Difference between AD and AWF models in terms of streamwise velocity and turbulence intensity for a square wind farm layout with $8D$ spacing. AWF models differ in horizontal spacing ($\Delta$) and CFD grid spacing ($\delta$).

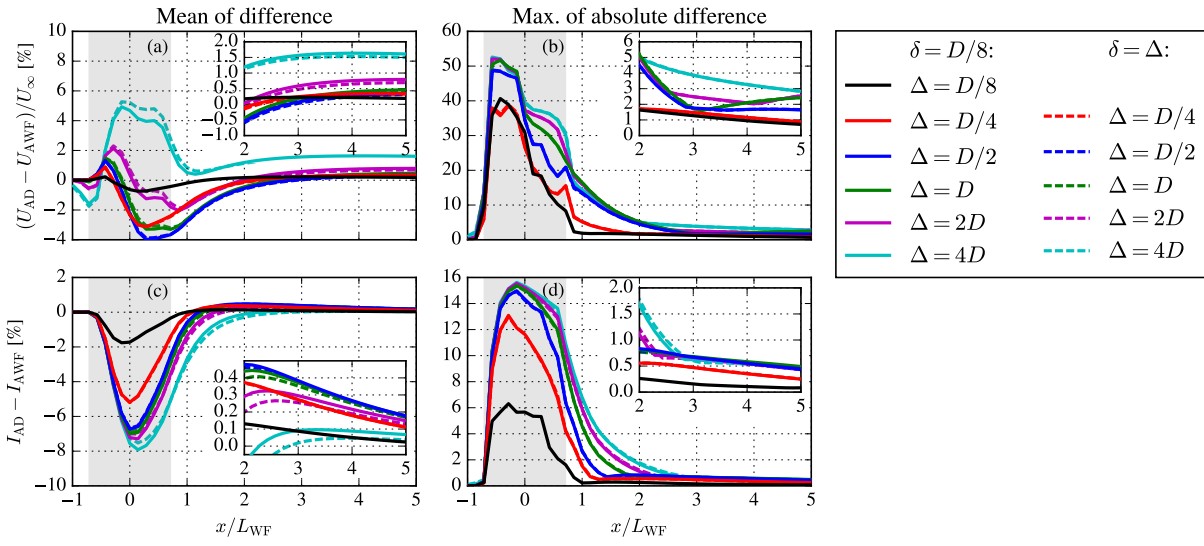

**Figure 11.** Same as Fig.10 but for a smaller wind turbine spacing of $4D$.

in streamwise velocity and turbulence intensity are in the order of 6% and 1%, respectively, but often only occur locally. A similar grid study has been performed for the same wind farm layout but with a wind turbine spacing of $4D$; the results are depicted in Fig. 11. In general, the differences between the AWF and AD models are larger compared to the lower density wind

farm. However, the mean error in streamwise velocity for the chosen setup using $\Delta = 2D$ and $\Delta = \delta$ is still only $\pm 0.7\%$ for $x > 2L$.

## 4.2 Model validation

### 4.2.1 Double wind farm case: Sheringham Shoal and Dudgeon

The effect of Sheringham Shoal on Dudgeon for a wind direction of $235 \pm 2.5°$ and a wind speed of $8 \pm 0.5$ m s$^{-1}$ is simulated with RANS, TurbOPark and the WRF model. Three RANS setups are used; two employ either all ADs or all AWF models for both wind farms, while the third uses ADs for the downstream wind farm (Dudgeon) and an AWF model for the upstream wind farm (Sheringham Shoal). The horizontal canopy spacing of the AWF models is set to $2D$. The RANS models are simulated for a wind speed of 8 m s$^{-1}$ and every 5° between 220–250° and post processed by a Gaussian filter of 5°. PyWake's TurbOPark implementation follows the original formulation by Nygaard et al. (2020) (as close as possible) and a revised setup where the ground model is switched off and a wake expansion coefficient of 0.06 is employed instead of 0.04. TurbOPark is used for the same wind speeds as the RANS setups and for every 1°, and also post processed using the same Gaussian filter and a subsequent linear average between $235 \pm 2.5°$. The Gaussian filter is applied to represent the uncertainty of the measured wind direction (Gaumond et al., 2014; van der Laan et al., 2015b), a mathematical description can be found in Antonini et al. (2019). The results of the double wind farm case are depicted in Fig. 12, in terms of horizontal wind speed contours Figs. 12a-e, wake magnitude, Fig. 12f, and shape, Fig. 12g, the latter two are obtained from the wind turbine power of the front row of Dudgeon using the wind turbine power curve. The wake shape represents the wind farm wake velocity deficit normalized by its mean value, while the wake magnitude refers to the wind farm wake velocity deficit normalized by the freestream. A wake shape is used for validation because the SCADA measurements of the front row turbines are both used to determine the freestream wind speed, $U_{\mathrm{ref}}$, and to measure the upstream wind farm wake due to the lack of freestream measurements, as discussed in Sect. 2. Figure 12h also shows results of the wake of Sheringham Shoal extracted along a 145–325° transect (perpendicular to the main wind direction of 235°), at 1.4 km upstream of Dudgeon. These results are used in order to compare results of all RANS models and TurbOPark with those using the WRF model because it is not possible to use the wind turbine power output of the front row of wind turbines from the WRF simulation to extract the wake effect of Sheringham Shoal due to the large horizontal resolution. The upstream distance of 1.4 km is chosen to avoid cells that include a wind turbine in the WRF model setup. In addition, $U_{\infty}$ for the WRF model represents the wind speed from a simulation without turbines, while $U_{\infty}$ is the actual inflow wind speed in the RANS and TurbOPark models. Finally, results from simulating the Dudgeon wind farm alone using RANS-AD are also depicted in Fig. 12e, which shows that wind farm induction is in the order of 1% of the freestream and much smaller than the effect of Sheringham Shoal on Dudgeon predicted by the RANS and TurbOPark simulations that also include Sheringham Shoal.

The contours plots in Figs. 12a-e are rotated to better visualize the incoming flow for the front row wind turbines and transect, for which the results are depicted in Figs. 12f-h. Figs. 12b-d show comparable wind farm wakes between the RANS models. The horizontal wind speed flow fields from the WRF model results (Fig. 12e) are quite different compared to those

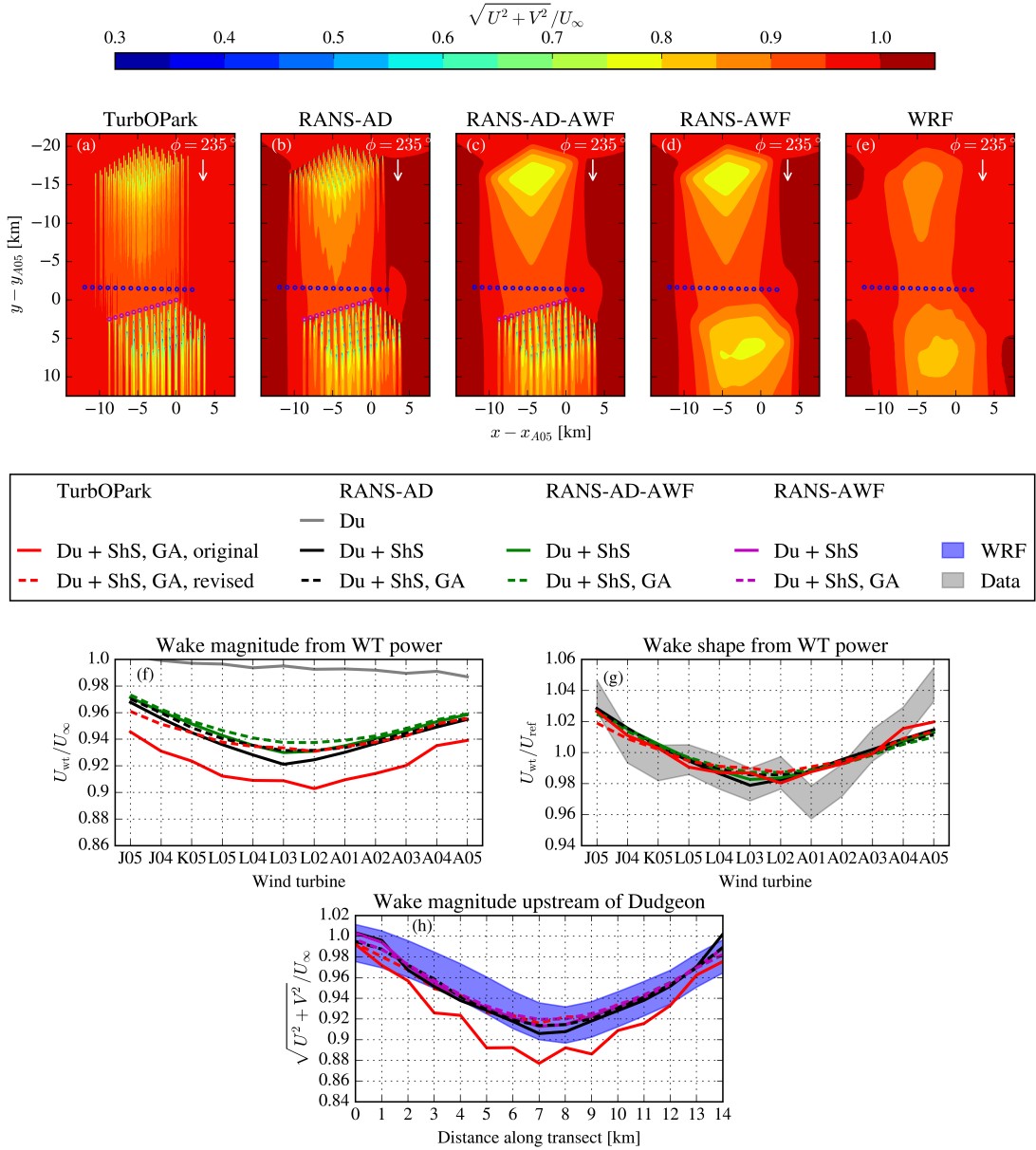

**Figure 12.** Double wind farm case: Effect of Sheringham Shoal (ShS) on Dudgeon (Du) for a wind direction of $235 \pm 2.5°$. The effect is shown in terms of velocity deficit magnitude **(f)** and shape **(g)** based on the Southern front row wind turbine (WT) power of Dudgeon (purple markers), and the velocity deficit magnitude extracted at a transect upstream of Dudgeon **(h)** (blue markers). Gaussian averaging (GA) is applied for the results shown as dashes lines. Contours of horizontal wind speed at $z = 102$ m, are shown in **(a–e)** for TurbOPark (revised) **(a)**, RANS-AD **(b)**, RANS-AD-AWF **(c)**, RANS-AWF **(d)** and WRF **(e)**. The contours reflect a single wind direction of $235°$ for the RANS models and TurbOPark, while the WRF contours are obtained from $235 \pm 2.5°$.

obtained from the RANS models. For example, the WRF model simulation lacks regions of increased wind speed at the sides of Sheringham Shoal, although the flow in between the wind farms is more similar. The horizontal wind speed deficits of TurbOPark (using the revised setup) shown in Fig. 12a have distinct single wind turbine wakes, due to the applied superposition method because the depicted contours in Fig. 12a reflect a single wind direction of 235°.

There is a maximum of 0.9% difference between the RANS-AD and RANS-AD-AWF models in terms of the equivalent wind speed extracted from the first row wind turbines (with respect to the freestream wind speed). Note that it is not possible to perform this exercise with the RANS-AWF simulations, since both wind farms are represented by an AWF model. This difference in wind speed is smaller along the transect and all three RANS models predict a wake magnitude in the range of the WRF model results, as shown in Fig. 12e. The results from the RANS-AD-AWF and RANS-AWF models are nearly identical

(Fig. 12h) because the upstream effect on Dudgeon is small at the transect. TurbOPark in its original formulation predicts stronger wind farm wake effects compared to RANS for the front row of Dudgeon (Fig. 12f) and compared to the WRF model results for the transect (Fig. 12h). However, in terms of wake shape (Fig. 12g), all models compare relatively well with the SCADA measurements, which shows the difficulty of validating the models with SCADA without freestream measurements. TurbOPark in its revised setup predicts much better results compared to those of RANS and the WRF model, partially due to

removing the mirror plane for the wake deficits. Whilst not relevant in the near wind farm wake, the mirror deficits become active at large distances from the turbine of origin, when hitting Dudgeon.

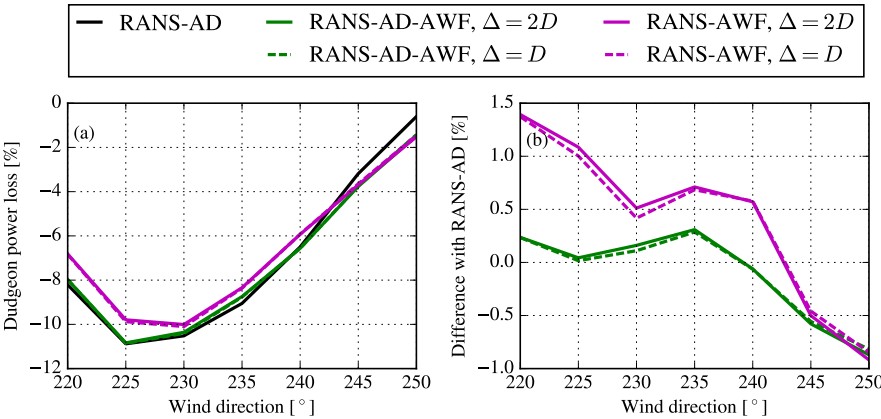

**Figure 13.** Double wind farm case: Effect of Sheringham Shoal on Dudgeon in terms of wind farm power loss **(a)** using the three RANS models and two horizontal grid resolutions for the AWF model. The difference with the RANS-AD model is shown **(b)**.

     The wind farm power loss of Dudgeon due the presence of Sheringham Shoal is depicted in Fig. 13, where results of the three RANS methods are shown as function of wind direction in Fig. 13a. In addition, the RANS-AD-AWF and RANS-AWF models are also simulated with a finer horizontal resolution of $\Delta = 1D$. Furthermore, the difference with respect to the RANS-

550 AD simulation is plotted in Fig. 13b. When the downstream wind farm is also modeled as an AWF (RANS-AWF), then the difference with the RANS-AD model results is generally larger (up to 1.5%) except for the wind directions of 245 and 250°. The error in the RANS-AWF model is not strongly related to the grid size, since refining the horizontal spacing by a factor of

two only changes the results of the RANS-AWF by a maximum of 0.1%. We suspect that the main source of error is the fact that the AWF model is controlled as an entire object instead of individual wind turbine control, as performed for the RANS-AD model. When the downstream wind farm is operating in a half wind farm wake situation, mostly applicable for the wind directions 220–225 and 245–250°, then the entire AWF is still affected if the change in the AWF volume averaged wind speed or wind direction changes the wind farm thrust coefficient, which would not be the case when the downstream wind farm is represented by ADs. If the latter is important, then one could consider splitting each wind farm into several AWF models or simply model the wind farm of interest with ADs following the RANS-AD-AWF approach.

### 4.2.2  Wind farm cluster

The wind farm cluster consisting of the three wind farms Dudgeon, Sheringham Shoal and Race Bank is simulated with RANS-AD-AWF, where Dudgeon is represented by ADs and the other two wind farms are modeled with the AWF model. A range of wind directions between 190 and 300° is simulated for every 5° using a wind speed of 8 m s$^{-1}$. The wind turbine power of the first row of Dudgeon is used to calculate the effective wind speed and the results are Gaussian averaged using the same standard deviation as employed in Sect. 4.2.1. The RANS simulations are compared with the SCADA measurements for the Southern and Western front rows of Dudgeon in Figs. 14 and 15, respectively. The results are normalized by $U_{\mathrm{ref}}$, which represents the mean row wind speed that is used to determine the freestream in the SCADA measurements due to the lack of freestream measurements. Hence, the wake shape is only validated, while the wake magnitude is not, as also discussed in Sect.4.2.1. Figures 14 and 15 indicate that the RANS simulations capture the overall trend of the wind farm wake shape as function of wind directions. For the Southern row of Dudgeon, the measurements indicated a more pronounced wind farm wake shape, best visible by the difference in simulation and measurements at the corner wind turbines J05 and A05 for the wind directions between 205 and 245°, and 235 and 250°, respectively. This could indicate stronger wake effects in the measurements, possibly due to the effect of near stable conditions, which is not accounted for in the simulations.

The RANS results of the Western row in Fig. 15 are more difficult to interpret because the Western row width is not large enough to capture the entire wind farm wake of Race Bank, which leads to flat wake shapes when the Western front row of Dudgeon is operating in the full wind farm wake of Race Bank, as seen for the wind direction of 265°.

### 4.3  Computational effort

The main motivation for the AWF model is the reduction in computational effort compared to using ADs. Table 3 provides an overview of the CFD grid sizes and computational effort per flow case (i.e. a single inflow wind speed and direction) for running one, two and three wind farms with different combinations of wind farms modeled as AD and/or AWFs. The underlying CFD solver in PyWakeEllipSys, EllipSys3D, is a flow solver based on a block structured grids, which uses a Message Passing Interface (MPI) for communication. In the present work, we use block (edge) sizes of 32 and 64 corresponding to $32^3$ and $64^3$ cells per block, respectively. The largest part of the computational effort is used to solve for the pressure correction equation. A multi-grid algorithm is applied to solve the pressure correction equation and the coarsest level is made parallel on a single node using shared memory MPI. The most recent version of EllipSys3D is expected to scale efficiently and be able to run with

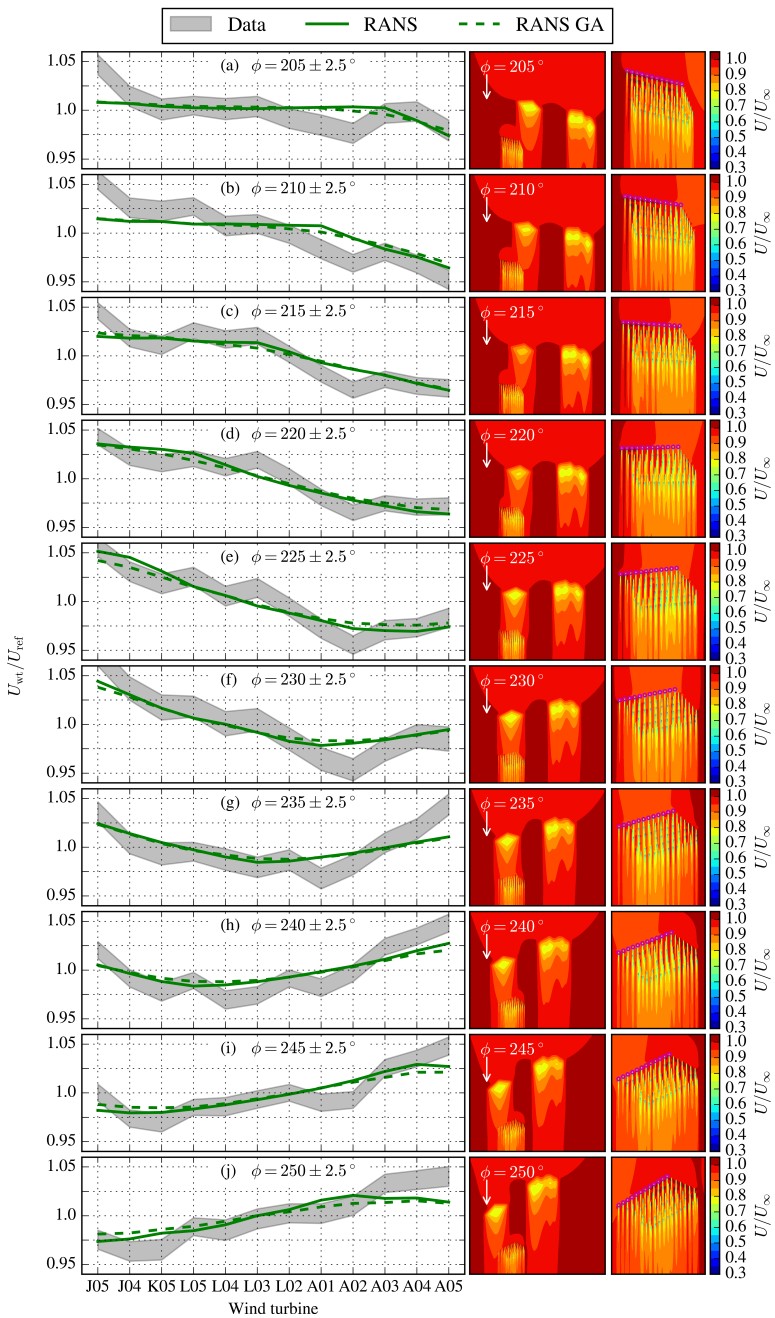

**Figure 14.** Measured and simulated effect of the upstream wind farms on the Southern wind turbine row of Dudgeon in terms of wake shape for different wind directions **(a-j)**. Results are normalized by the row-averaged wind speed. Contours of the streamwise velocity at the reference height, normalized by the inflow wind speed, are shown for each wind direction case, including zoomed plot around Dudgeon.

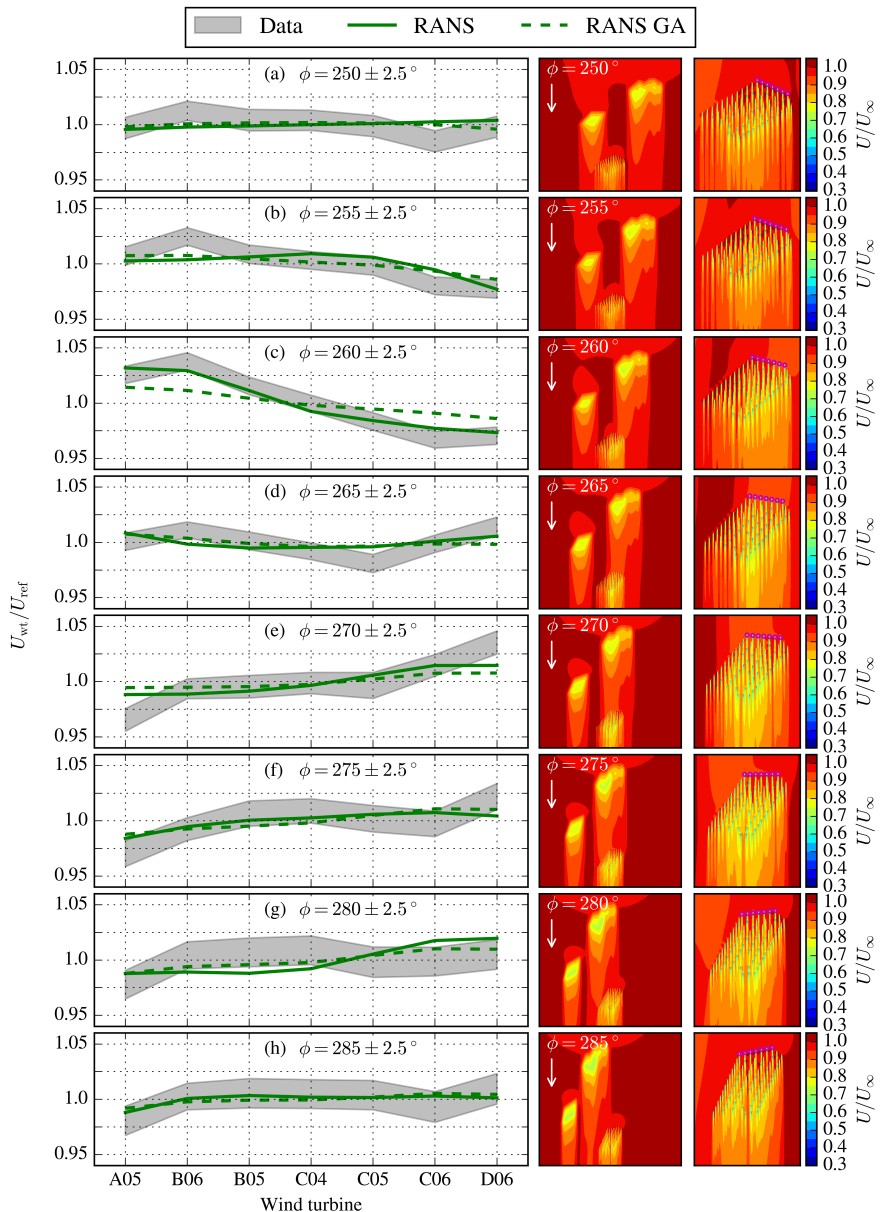

**Figure 15.** Same as Fig. 14 for the Western row of Dudgeon for different wind directions **(a-h)**.

one block per processor for grids with 100000 blocks of $64^3$. Four shared memory CPUs are used for the wind farm cluster simulations using three wind farms and only ADs due to the large number of blocks (4550), while the other simulations are not using shared memory CPUs. The AD model in EllipSys3D is recently made more scalable by distributing the same number of

ADs to each CPU as much as possible; initial tests have shown good scalability up to 1000 ADs (van der Laan et al., 2023). Further optimizing the computational effort and scalability of large wind cluster simulations using ADs is planned in the future.

The RANS-AD grids for multiple wind farms could be reduced by using more complex grid topologies, where each wind farm is situated in a refined region instead of using a single large refined area that includes all wind farms. However, it is more challenging to generate such a grid topology compared to box type domains. Furthermore, it should be noted that the simulations are performed on an in-house shared computer cluster. Hence, the listed CPU hours and wall clock time in Table 3 should be interpreted as indicative. The computer cluster consists of 516 computational nodes, each one equipped with two sixteen-core AMD EPYC 7351 2.9 GHz processors and 128 GB RAM of memory (Technical University of Denmark, 2019).

| Case | Method | Cells [million] | Block size | Blocks & CPUs | Wall clock | CPU hours | CPU hours incl. Steps 1a and 1b |
|------|--------|-----------------|------------|---------------|------------|-----------|---------------------------------|
| Single AWF $C_{T,\mathrm{wf}}$ and $C_{P,\mathrm{wf}}$ generation (Step 1a) | | | | | | | |
| ShS | RANS-AD | 115 | 64 | 437 | 0.20 | 88 | - |
| Rb | RANS-AD | 163 | 64 | 621 | 0.25 | 153 | - |
| Du | RANS-AD | 115 | 64 | 437 | 0.22 | 102 | - |
| Single AWF force calibration (Step 1b) | | | | | | | |
| Shs | RANS-AWF | 1.05 | 32 | 32 | 0.0080 | 0.25 | - |
| Rb | RANS-AWF | 1.31 | 32 | 40 | 0.0072 | 0.28 | - |
| Du | RANS-AWF | 1.31 | 32 | 40 | 0.0066 | 0.26 | - |
| Double wind farm simulations | | | | | | | |
| Du + ShS | RANS-AD | 297 | 64 | 1134 | 0.72 | 819 | - |
| Du + ShS | RANS-AD-AWF | 110 (-63.0%) | 64 | 420 | 0.50 (-32.6%) | 204 (-75.1%) | 385 (-53.0%) |
| Du + ShS | RANS-AWF | 3.15 (-98.9%) | 32 | 96 | 0.027 (-96.7%) | 2.30 (-99.7%) | 421 (-48.6%) |
| Wind farm cluster simulations | | | | | | | |
| Du + ShS + RB | RANS-AD | 1193 | 64 | 4550 | 1.53 | 6958 | - |
| Du + ShS + RB | RANS-AD-AWF | 252 (-78.9%) | 64 | 960 | 0.56 (-63.4%) | 538 (-92.3%) | 1282 (-80.1%) |
| Du + ShS + RB | RANS-AWF | 6.55 (-99.5%) | 32 | 200 | 0.026 (-98.3%) | 2.23 (-99.9%) | 1206 (-82.7%) |

**Table 3.** Grid size and computational effort of RANS wind farm (cluster) simulations including Dudgeon (Du), Sheringham Shoal (ShS) and Race Bank (RB). Percentage between brackets reflects the reduction when using AWF models with respect to only using ADs. CPU and run hours are listed per flow case.

When the investigated wind farm cluster is simulated with Dudgeon as ADs and the other wind farms as AWF models, 76% of cells can be saved compared to modeling the wind farm cluster with only ADs due to a reduction in horizontal spacing outside the Dudgeon wind farm area, as listed in Tab. 3. For the double wind case, 63% of the cells can be saved and the number of CPU hours per flow case is reduced by 75%. When the wind farm cluster consisting of three wind farms is solely modeled by AWF models, then the grid size is reduced by 99.5% and a flow case can be solved in about 1.5 min. using 200 CPUs, which is a reduction of 99.9% in terms of CPU hours. However, each AWF model requires information of a wind farm

thrust coefficient, $C_{T,\mathrm{wf}}$, and the wind farm power coefficient (if the AWF model is used to predict wind farm power), $C_{P,\mathrm{wf}}$, as function of the AWF volume averaged wind speed and wind direction. If this input is derived from a set of RANS simulations using ADs, as performed in the present work, then the computational effort will be dominated by this step. For example, Tab. 3 shows that the Sheringham Shoal wind farm modeled by ADs takes about 0.20 h per flow case or 88 CPU hours using a grid of 115 million cells. The wind farm cluster modeled by three AWFs requires about 1206 CPU hours per flow case when including the wind farm precursor steps, which is still a reduction of 83% compared to simulating all wind farms as ADs (6958 CPU hours per flow case), as shown by the last column of Tab. 3. One could attempt to use a simplified wake model to determine $C_{T,\mathrm{wf}}$ and $C_{P,\mathrm{wf}}$ but one would risk a loss of accuracy. An initial study shows that is possible to predict $C_{T,\mathrm{wf}}$ and $C_{P,\mathrm{wf}}$ within a mean absolute error of around 1% when using a RANS-based surrogate wind farm model (van der Laan et al., 2023).

## 5  Conclusions

A RANS-based wind farm parameterization, the AWF model, is proposed. It uses a wind farm thrust force as a momentum sink similar to a forest canopy model. The AWF model can be used as an obstacle model to a downstream wind farm of interest represented by ADs or it can be used to estimate wind farm power when the downstream wind farm is also modeled as an AWF. The AWF model simulates wind farm interaction by a calibrated controller of wind farm thrust and power as function of the wind farm volume averaged wind direction and wind speed.

When the horizontal spacing in the AWF model is refined, the wind farm flow approaches the results from a RANS-AD wind farm simulation. This is achieved by calibrating the thrust force magnitude with precursor RANS-AD wind farm simulations and employing a horizontal thrust force distribution in the form of wind turbine density using a superposition of the wind turbine coordinates represented by two-dimensional Gaussian functions. The verification study showed that the Gaussian superposition method solves the problem of artificial wind farm wake effects that can occur when the number of wind turbines are binned for large horizontal cells, as current wind farm parametrizations implemented in numerical weather models, such as the WRF model, do.

A new atmospheric inflow model is introduced that is potentially more suited for wind farm cluster simulations because it does not rely on an ABL height set by a global turbulence length scale limiter that can result in nonphysical wind farm wakes. The proposed inflow model relies on a prescribed analytical temperature profile including an inversion height and inversion strength, while a temperature equation is not solved for in order to maintain a steady-state inflow model. The model is shown to be dependent on three non-dimensional numbers. The new inflow model does not (and is not expected to) solve the problem of numerical instabilities related to the low eddy viscosity region above the ABL in combination with large horizontal domains associated with wind farm clusters. The problem is mitigated in the present work by using a relatively low domain height, which can introduce additional numerical blockage, thereby negatively influencing the predictions. An alternative solution needs to be found in the future to allow the more preferred, taller domain heights. In addition, the new ABL model requires more validation with measurements.

The proposed RANS-AWF and inflow models are employed to simulate two neighboring wind farms and a cluster consisting of three wind farms, where either one of the wind farms is modeled by ADs and the remaining wind farms are represented by AWF models (RANS-AD-AWF) or all wind farm are AWF models (RANS-AWF). The results for the double wind farm case are compared with TurbOPark, WRF and RANS-AD simulations (for the latter all wind turbines are modeled by ADs) using the wind speed derived from the front row turbines of the downstream wind farm and the horizontal wind speed extracted from a transect 1.4 km upstream of the wind farm farthest downstream. The latter is performed because the chosen resolution for the WRF model simulations was not sufficient to resolve the front row wind turbine wind speed. While the overall horizontal wind speed at the wind farms in the WRF model simulations is quite different with respect to the RANS results, the horizontal wind speed at the transect from the WRF results compares well with those from the RANS-AD, RANS-AD-AWF and RANS-AWF models. In addition, the front row wind speed in the RANS-AD-AWF only deviated by 0.9% compared to the RANS-AD results with respect to the freestream. The original formulation of TurbOPark shows stronger wind farm wake effects compared to the other models but its result in terms of wind farm wake shape compared well with all models. This indicates that a comparison in terms of wind farm wake shape should not be the only type of validation. A revised TurbOPark setup, where the ground model is switched off and a larger wake expansion coefficient of 0.06 is used, predicts much better results compared to the higher fidelity models. Unfortunately, the RANS-AD-AWF simulations of the wind farm cluster could only be validated with the shape of the front row wind speed because the SCADA measurements of this row are used to both measure the wake of the upstream wind farm and to determine the freestream conditions due to a lack of concurrent freestream measurements. The trends of the RANS-AD-AWF simulation results of the upstream wind farm wake shapes compare reasonably well with the results from the SCADA, although the measured shapes indicate stronger wind farm wake effects possibly due to near stable conditions that were not filtered out from the SCADA in order to maintain a large enough data set. More validation of both the AWF model and prescribed temperature inflow model are required. In addition, we need SCADA with concurrent inflow measurements in order to validate the magnitude of wind farm wakes and its impact on neighboring wind farms.

With the AWF model, one can simulate large wind farm clusters with RANS; the wind farm cluster validation case showed a reduction of 92.3% and 99.9% in CPU hours when two of three wind farms or all wind farms are represented by AWF models instead of using ADs, respectively, when the input wind farm thrust and power coefficient are known. If the wind farm power and thrust coefficients are calculated from a RANS-AD simulation of each wind farm, as performed in the present work, then the reduction in computational effort is 82.7% when the wind farm cluster validation case is solely modeled by AWF models. Simpler and faster models that can generate the AWF input are currently investigated in a follow up work (van der Laan et al., 2023). This also enables a larger wind farm cluster to be efficiently simulated with the RANS-AWF methodology in future work. Such a study would also benefit from a comparison with WRF model results to further investigate the effect of neglecting the mesoscales in the RANS-AWF approach.

The AWF model can be used to further develop wind farm parametrizations in WRF; this can be achieved by implementing the two-dimensional Gaussian superposition method of the wind turbine density, and by using a wind farm drag coefficient that is dependent on the wind direction instead of using the wind turbine thrust curves. Idealized WRF simulations could be used to compare the resulting wind farm wakes with RANS-AWF simulation results.

The current implementation of the AWF model does not include any sources in the turbulence transport equations because the employed horizontal spacing of $2D$ turned out to be sufficiently fine. If larger horizontal cell size are desired one could investigate the use of additional turbulence-related source terms.

## Appendix A: Similarity of the prescribed temperature inflow model

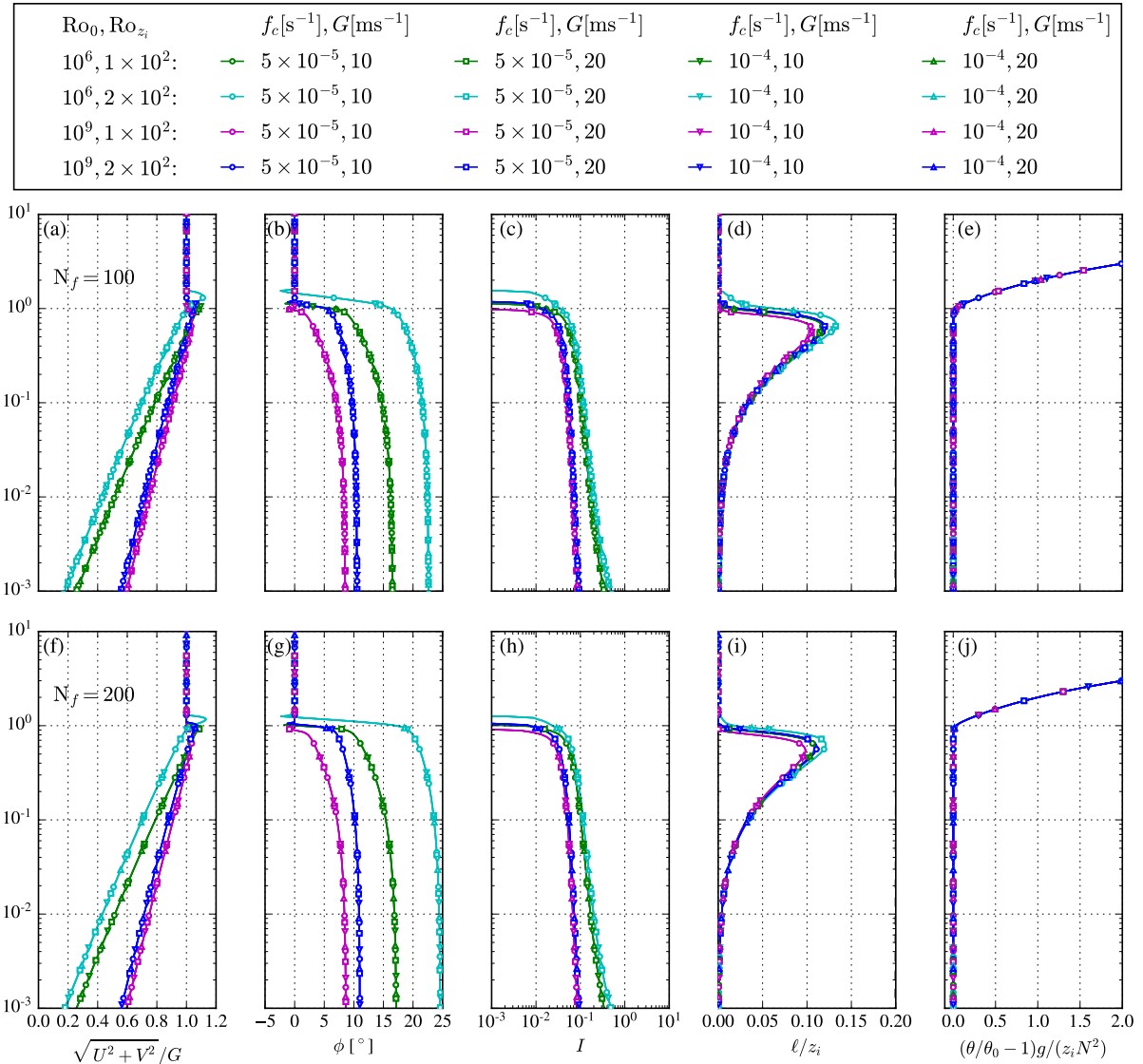

**Figure A1.** Similarity of prescribed temperature inflow model for $N_f = 100$ **(a–e)** and $N_f = 200$ **(f–j)**.

In previous work (van der Laan et al., 2020), it was shown that the ABL model of Apsley and Castro (1997), including Coriolis forces, follows a Rossby similarity; here four dimensional input parameters (namely the geostrophic wind speed $G$, roughness length $z_0$, Coriolis parameter $f_c$, and $\ell_{\max}$) can be reduced to two Rossby numbers, $\mathrm{Ro}_0 \equiv G/(|f_c|z_0)$ and $\mathrm{Ro}_\ell \equiv G/(|f_c|\ell_{\max})$. We typically generate a library of normalized ABL profiles for all possible solutions, which can then be used to find values for $G$ and $\ell_{\max}$ in order to get a desired wind speed and turbulence intensity at a reference height for a fixed roughness length and Coriolis parameter. The new prescribed temperature ABL model from Sect. 3.1.3 depends on five dimensional parameters; $G$, $z_0$, $f_c$, $z_i$, $\partial\theta/\partial z|_c$; however, the normalized profiles only depend on three non-dimensional numbers:

$$\mathrm{Ro}_0 \equiv \frac{G}{|f_c|z_0}, \qquad \mathrm{Ro}_{z_i} \equiv \frac{G}{|f_c|z_i}, \qquad \mathrm{N}_f \equiv \frac{N}{|f_c|} = \frac{1}{|f_c|}\sqrt{\frac{g}{\theta_0}\frac{\partial\theta}{\partial z}\bigg|_c} \tag{A1}$$

Here, $\mathrm{Ro}_0$ is the surface Rossby number, $\mathrm{Ro}_{z_i}$ is a Rossby number based on the inversion height and $\mathrm{N}_f$ is dimensionless number based on the ratio of the Brunt–Väisälä frequency and the Coriolis parameter, which is also referred to as the Zilitinkevich number (Esau, 2004; Kelly et al., 2019; Liu et al., 2021). The similarity is numerically proven by the collapse of simulations results for different values of $G$ and $f_c$, as depicted in Fig. A1. The numerical setup is similar to van der Laan et al. (2020), but we have chosen a smaller time step set to $1/N$ instead of $1/|f_c|$ in order to maintain numerical convergence. From the figure one can note that $\mathrm{N}_f$ mostly changes the effective sharpness of the inversion, as seen in $\phi(z)$, as well as the ABL turning angle $\phi(z = z_0)$; the latter effect is more pronounced for larger $z_i/z_0$. It is also possible to use an alternative dimensionless number for quantifying the inversion strength, using an ABL-wide bulk Richardson number (instead of $\mathrm{N}_f$):

$$\mathrm{Ri} \equiv \frac{z_i^2 N^2}{G^2} = \frac{z_i^2}{G^2}\frac{g}{\theta_0}\frac{\partial\theta}{\partial z}\bigg|_c. \tag{A2}$$

## Appendix B: Comparison and challenges of RANS inflow models applied to the wind farm cluster validation case

A new atmospheric inflow model is proposed in Sect. 3.1.3 and used to perform the RANS wind farm cluster simulations in Sect. 4. The new inflow model employs a prescribed temperature profile with an inversion that sets the ABL height. Previous work employed an ABL inflow model based on a global turbulence length scale limiter, $\ell_{\max}$ (Apsley and Castro, 1997; van der Laan and Sørensen, 2017b; van der Laan et al., 2017). For a low value of $\ell_{\max}$, a lower ABL height is obtained. A major issue with the global turbulence length scale limiter is that it can also limit the wind farm wake turbulence length scales that could lead to a nonphysical slow wake recovery. This issue was mitigated by switching off the global length scale limiter in regions where high velocity gradients are present (van der Laan and Sørensen, 2017b), although it is unclear if this ad-hoc solution is sufficient for wind farm cluster simulations. Another challenge is that for low ABL heights, one can obtain numerical oscillations, which both the new model and the model based on $\ell_{\max}$ can suffer from. This section is meant to illustrate these issues and to show the difference between the inflow models when applied to the wind farm cluster validation case.

Table B1 lists the input and derived parameters of a shallow and a tall ABL inflow using the new prescribed temperature ABL model and the ABL model based on $\ell_{\max}$. The tall ABL case is the same as used through out the article (Sects. 3.1.3 and

| Case | Model | Input parameters | | | | | | Derived parameters | | |
|---|---|---|---|---|---|---|---|---|---|---|
| | | $I_{\mathrm{amb}}$ | $z_i$ | $\theta_0$ | $\frac{\partial\theta}{\partial z}\vert_c$ | $\frac{z_T}{z_i}$ | $z_0$ | $G$ | $z_0$ | $\ell_{\max}$ |
| | | [-] | [m] | [K] | [K m$^{-1}$] | [-] | [m] | [m s$^{-1}$] | [m] | [m] |
| Tall ABL | Prescribed temp. | $10^{-5}$ | 1000 | 285 | $5\times10^{-3}$ | 0.2 | - | 8.50 | $3.25\times10^{-5}$ | - |
| Tall ABL | $\ell_{\max}$ | $10^{-6}$ | - | - | - | - | $5\times10^{-5}$ | 8.62 | - | 53.3 |
| Shallow ABL | Prescribed temp. | $10^{-5}$ | 300 | 285 | $5\times10^{-3}$ | 0.4 | - | 8.54 | $1.77\times10^{-4}$ | - |
| Shallow ABL | $\ell_{\max}$ | $10^{-6}$ | - | - | - | - | $5\times10^{-3}$ | 8.93 | - | 7.62 |

**Table B1.** Summary of input and derived parameters for the ABL inflow models, both using $I_{\mathrm{ref}} = 0.044$, $U_{\mathrm{ref}} = 8$ m s$^{-1}$, $z_{\mathrm{ref}} = 102$ m and $f_c = 1.168\times10^{-4}$ s$^{-1}$.

4), while the shallow ABL is an additional case meant to illustrate the challenges with the two inflow models. For the $\ell_{\max}$ ABL inflow model, the derived values for $G$ and $\ell_{\max}$ are found by interpolating a pre-calculated library of ABL profiles (van der Laan et al., 2020) for given $z_0$, $f_c$, $I_{\mathrm{ref}}$ and $U_{\mathrm{ref}}$ values. The prescribed temperature model uses an optimizer for $G$ and $z_0$ instead, as discussed in Sect. 3.1.3. The tall ABL profile from the $\ell_{\max}$ ABL inflow model uses a similar roughness length ($z_0 = 5\times10^{-5}$ m) as was calculated for the inflow model based on the prescribed temperature (Tab. 2), which results in a relative large value of $\ell_{\max}$, namely 53.3 m. The shallow ABL profile of the prescribed temperature ABL model is generated with a lower $z_i$ (300 m) resulting in a larger derived $z_0$ (Tab. B1). In addition, the value of $z_T/z_i$ as used in the prescribed temperature ABL model is increased for the shallow ABL to avoid an inflection of the wind speed profile below the super geostrophic jet. To lower the ABL height for model based on $\ell_{\max}$, one can lower $z_0$ or $I_{\mathrm{ref}}$; we choose to set $z_0 = 5\times10^{-3}$ m. This results in a derived value of $\ell_{\max}$ of only 7.62 m. Finally, the ABL model based on $\ell_{\max}$ is employed with a lower ambient turbulence intensity value, $I_{\mathrm{amb}}$, compared to the prescribed temperature model because the $\ell_{\max}$ ABL model is more sensitive to this parameter. The chosen values for each ABL model are sufficient to not influence the numerical solution (van der Laan et al., 2020). The results of all profiles are depicted in Fig. B1. Here it is clear that over the rotor swept area, the profiles of wind speed, direction, and normalized turbulence intensity resulting from the prescribed temperature model are similar to those resulting from the $\ell_{\max}$ model for the tall ABL case. The main difference can be found in the turbulence model length scale (Fig. B1d), where the prescribed temperature ABL model predicts larger length scales over the rotor swept area compared to the $\ell_{\max}$ based ABL model. For taller altitudes, the models produce quite different ABL solutions probably because the prescribed temperature model sets an explicit inversion strength while the ABL model based on $\ell_{\max}$ calculates this implicitly. For the shallow ABL case, both models predict very different profiles. This is because the model based on $\ell_{\max}$ represents stable conditions for low values of $\ell_{\max}$ while the prescribed temperature ABL model reflects a conventionally neutral ABL.

The four inflow profiles listed in Tab. B1 are applied to the wind farm cluster validation case using the RANS-AD-AWF model, for a wind direction and wind speed of 235° and 10 m s$^{-1}$, respectively. The results of these simulations in terms of horizontal wind speed contours at hub height ($z = 102$ m) are depicted in Fig. B2. Here, the prescribed temperature model for the tall ABL case (Fig. B2a) is same as shown earlier in Fig. 14g and its results are similar as the results of the inflow model based on $\ell_{\max}$ (Fig. B2b) for the tall ABL case. This is because a large ABL of about 1 km is set by either using a large

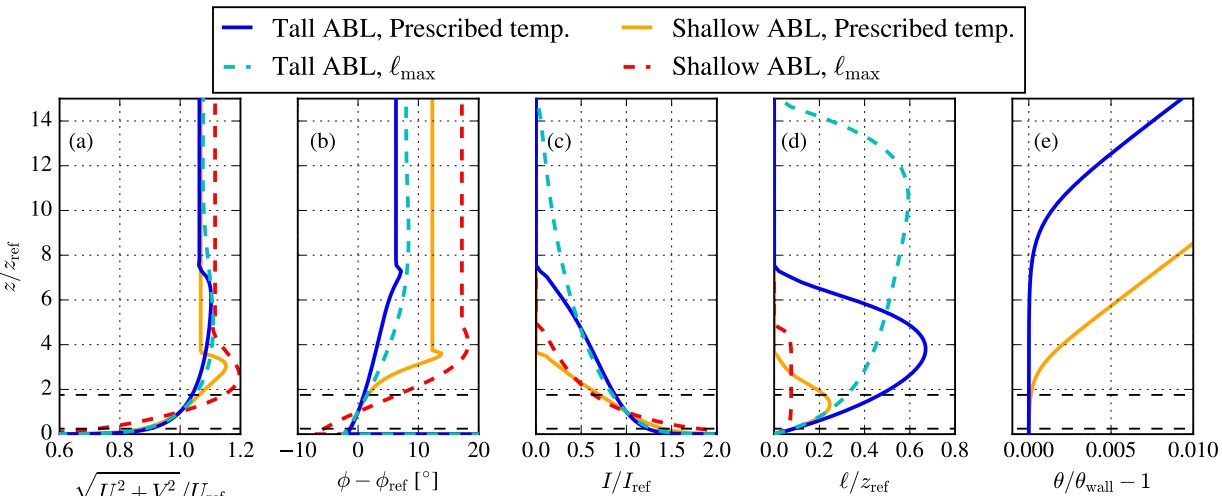

**Figure B1.** Atmospheric inflow model precursor results employing a global turbulence length scale limiter, $\ell_{max}$ compared to the prescribed temperature ABL model for shallow and tall ABL cases. Results are shown in terms of wind speed **(a)**, relative wind direction **(b)**, turbulence intensity **(c)**, turbulence length scale **(d)** and potential temperature **(e)**. Horizontal dashed black lines depict the swept rotor area of the 6 MW wind turbine.

$z_i$ or a large value of $\ell_{max}$. Figure B3 also shows a similar wake magnitude calculated by the two inflow models, where the difference between the models is in the order of 1%. The wake magnitude is based on the front row wind turbine power of Dudgeon, as performed in Sect. 4.2. Hence, the results presented in Sect. 4 would not have been significantly different if the inflow model based on $\ell_{max}$ was used instead of the new inflow model. The shallow ABL inflow profiles applied to the wind farm cluster do not lead to a converged result using the domain height, $L_z$, set to $10D_{ref}$, as employed for the tall ABL case. This is because the low eddy viscosity region above the ABL causes numerical instabilities when it is included in the wind farm cluster domain. The issue can be mitigated by lowering the domain height to exclude the low eddy viscosity region, as performed in previous work (van der Laan et al., 2017). Here, we apply two different values: $L_z/D_{ref} = 2.5D$ and $L_z/D_{ref} = 3D$ and the results are shown in Figs. B2c-B2e. It should be noted that these low domain heights are not desired because the increased numerical blockage causes artificial flow accelerations and reduced wake losses. When using $L_z/D_{ref} = 2.5D$ the shallow ABL simulations converge for both models (Figs. B2c and B2d). However, for $L_z/D_{ref} = 3D$ (Fig. B2e, prescribed temperature ABL model) and $L_z/D_{ref} = 4D$ (Fig. B2f, $\ell_{max}$ ABL model), the wind farm cluster simulation starts to produce numerical instabilities towards the outlet. It is clear that both ABL models cannot be used to simulate a wind farm cluster subjected to a shallow ABL inflow without lowering the domain height to an undesired small value. In addition, it is not fully determined if the new inflow model actually performs better than the model based on $\ell_{max}$ and more work is needed to improve the numerical behavior when considering low ABL heights. However, the prescribed temperature model offers a more physical

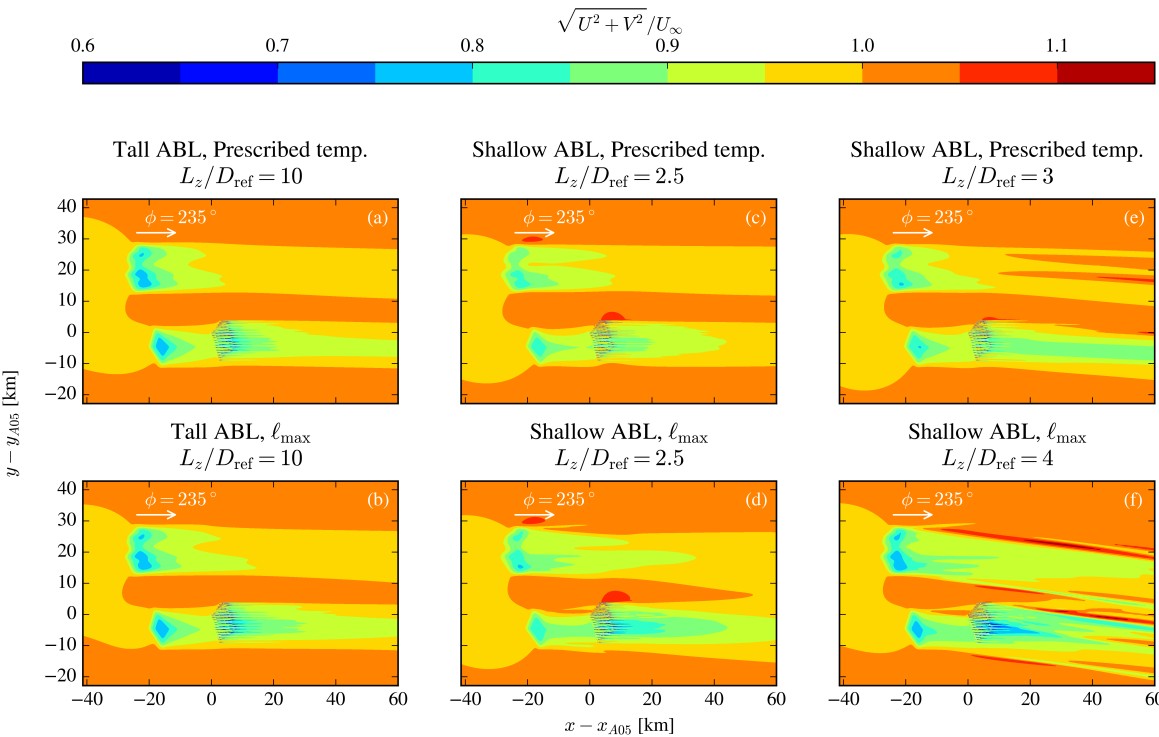

**Figure B2.** Contours of the streamwise velocity at the reference height of the validation case simulated with RANS-AD-AWF, for a wind direction of $235°$, for different inflow models and ABL heights.

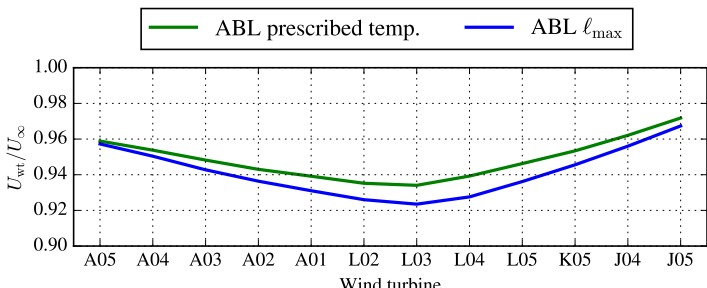

**Figure B3.** Wake magnitude of the validation case for a wind direction of $235°$ simulated with RANS-AD-AWF, for different inflow models using the tall ABL case.

method of setting an ABL height compared to using a global turbulence length scale limiter, and the new model also provides the possibility to explicitly set the inversion strength.

**Appendix C:  Numerical setup of TurbOPark (as implemented in PyWake)**

| Description | Original model | Revised model |
|---|---|---|
| Wind farm model | All2AllIterative | All2AllIterative |
| Wake deficit model | `TurboGaussianDeficit(`<br>`use_effective_ws=False,`<br>`use_effective_ti=False,`<br>`groundModel=Mirror(), A=0.04)` | `TurboGaussianDeficit(`<br>`use_effective_ws=False,`<br>`use_effective_ti=False,`<br>`groundModel=None, A=0.06)` |
| Blockage deficit model | `SelfSimilarityDeficit2020(,`<br>`groundModel=Mirror(),`<br>`superpositionModel=LinearSum())` | `SelfSimilarityDeficit2020(,`<br>`groundModel=Mirror(),`<br>`superpositionModel=LinearSum())` |
| Rotor velocity averaging model | `GaussianOverlapAvgModel(4, 3)` | `GaussianOverlapAvgModel(4, 3)` |
| Turbulence model | None | None |
| Wake superposition model | `SquaredSum()` | `SquaredSum()` |

**Table C1.** Summary of the original and revised TurbOPark setup in PyWake

An overview of the numerical setup of the original and revised TurbOPark model as implemented in PyWake (DTU Wind
Energy, 2022a) is provided in Table C1.

## Appendix D: Numerical setup of WRF simulations including wind farms

An overview of the numerical setup of the WRF simulations including wind farm is provided in Table D1.

**Table D1.** Summary table of WRF model configurations including initial and boundary conditions.

| Parameter | Value |
| --- | --- |
| **WRF version** | Version 3.7.1 with implemented EWP |
| **Spatial Settings** | |
| Domains | D1: 188 x 188 grid points, 9 km grid spacing, |
| | D2: 243 x 240 grid points, 3 km grid spacing |
| | D3: 216 x 216 grid points, 1 km grid spacing |
| Nesting strategy | one-way nested |
| Nudging strategy | spectral nudging[‡] applied above ABL |
| Vertical levels | 64 levels, dz≈25 m up to around 250 m |
| **Temporal Settings** | |
| Simulation length | blocks of 11 days including 24 h spin-up |
| Lateral boundary condition update interval | every 6 h |
| **Initial and boundary conditions** | |
| Forcing data | ERA5 (Hersbach et al., 2020) |
| Terrain data | GMTED2010 (30-arc-second, Danielson and Gesch, 2011) |
| Land cover data | ESA CCI-LC 2015 (Poulter et al., 2015) |
| Sea Surface Temperature | OSTIA (Donlon et al., 2012) |
| **Physics** | |
| Microphysics | WSM5 (Hong et al., 2004) |
| Radiation | RRTMG (Iacono et al., 2008) |
| Cumulus | Grell-Freitas[‡] (Grell and Freitas, 2014) |
| Land surface | Noah LSM (Tewari et al., 2004) |
| Planetary boundary layer scheme | MYNN2 (Nakanishi and Niino, 2006) |

[‡] only domain 1

*Author contributions.* MPVDL has drafted the article and produced the figures. MPVDL and OGS have proposed the idea of a wind farm model with a drag coefficient as function of wind direction. MPVDL proposed to represent the wind turbine positions in the AWF model as two-dimensional Gaussian functions. AMF suggested to use a standard deviation that is twice the horizontal AWF grid size and proposed to use of Eq. (3) for the volume average velocity vector of an AWF model; both ideas are based on lessons learned from actuator line modeling. MPVDL has proposed the inflow model using a prescribed temperature and MK proposed the analytical form of the prescribed temperature profile. CDB and KSS have prepared the SCADA measurements of Dudgeon. KSS proposed to revise the setup of the TurbOPark model

simulations. MI has performed and post-processed the WRF simulations. AMF has performed preliminary RANS-AD simulations of the double wind farm case including Dudgeon and Sheringham Shoal and AMF also implemented the TurbOPark model in PyWake. AP has post processed the NEWA data. NNS has implemented the necessary improvements to make the large wind farm cluster RANS simulations possible with EllipSys3D in an efficient and scalable way. PER has facilitated and supervised the research performed in this article. All authors contributed to the methodology and finalization of the paper.

*Competing interests.* The authors declare that they have no conflict of interest.

*Acknowledgements.* This is work is co-financed by Equinor ASA. In addition, the first author is inspired by Andrey Sogachev to model a wind farm as a forest canopy. Furthermore, the authors would like to thank Mads Mølgaard Pedersen for his support with the implementation of TurbOPark in Pywake. We also gratefully acknowledge the computational and data resources provided on the Sophia HPC Cluster at the Technical University of Denmark, DOI: 10.57940/FAFC-6M81. OGS was supported by the European Union's Horizon 2020 research and innovation program under grant agreement no. 861291 as part of the Train2Wind (https://www.train2wind.eu/) Marie Skłodowska-Curie Innovative Training Networks. AP's contribution is partly funded by Independent Research Fund Denmark through the 'Multi-scale Atmospheric Modeling Above the Seas' (MAMAS) project (nr. 0217-00055B)

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
