# Peer review of "A new RANS-based wind farm parametrization and inflow model for wind farm cluster modeling"

_Wind Energy Science, 2022_

## Author Comment (AC1)

**Reply to reviewers**

March 28, 2023

We would like to thank the three reviewers for their detailed feedback and suggestions to improve the article. In the next sections, the reviewers comments are copied and answered per comment (blue color). An additional document is provided that highlights all modifications with respect to the initial submitted version.

**Reviewer 1 (Gonzalo Pablo Navarro Diaz)**

**general comments**

Dear authors, the topic addressed in your work is very innovative and important for the wind energy industry. This is because you study how to simulate the wake interaction between neighboring farms and its impact on production, a problem that is emerging in current and future offshore wind farm projects. The development of wind farm parameterization was something missing in CFD-RANS, and you seem to have made the first step towards it. The comparison of this new development with the parameterization in WRF and the engineer wake model TurbOPark adds a good perspective with another way to simulate wind farms at a lower computational cost. The comparison with SCADA measurements has not been entirely satisfactory. The writing is clear and easy to follow. Below you can find my major and minor revisions.

**specific comments**

Major revisions (require changes in the results):

1. line 125- The lack of inflow velocity measurements, as well as non-availability of the SCADA data from neighboring farms make the comparison between simulations and SCADA on the first row of turbines very weak. I suggest deleting the comparison with SCADA from this work.

We agree that the lack of inflow measurements is major shortcoming of the model validation with the SCADA measurements. It is a problem that many researchers have to deal with and the lack of inflow measurements for wind farm wake cases is often the norm, not the exception, see for example Fischereit et al. (2022); van der Laan et al. (2015a). Even if we had the SCADA available of the upstream wind farm(s) then it is still difficult to define the freestream wind speed as a single value due to the spatial variability of wind speed caused by for example mesoscale and coastal effects. Despite these challenges, the current SCADA still provides a validation in terms of wake shape. In addition, when the entire upstream wind farm wake is captured by the front row of the downstream wind farm (most likely the case for the results in Fig. 13b) then one could say something about the magnitude of the wind farm wake, although we have not made any statement about this in the text. For these reasons, we decided to keep the comparison with the SCADA measurements and we believe that we have written the shortcomings of the validation clearly in text.

2. line 290- The efforts of your research group to couple the Apsley and Castro mixing length limitation methodology with the wake simulation using k-e-fp model is not mentioned in the paper (https://iopscience.iop.org/article/10.1088/1742-6596/1934/1/012012). Despite this, the new methodology that includes the inversion layer seems to solve the problem associated with the limitation of the mixing length in the wakes. Unfortunately, you do not investigate this advantage, for example by comparing both methodologies for a particular case. In the end, the reader will not know if this problem in the wake is solved and raises doubts about the validity of the results presented. Please, I would like to see more results related to this topic.

You raise a good point that we also thought about. We have now added results of the inflow model based on the global turbulence length scale limiter, as well as a shallow ABL case in a new Appendix (Appendix B), see end of this document.

3. line 335- The new methodology of simulating neutral ABL including the inversion layer looks promising. It would be very important that in figure 4 you add the precursor results using the previous methodology of mixing length limitation (https://iopscience.iop.org/article/10.1088/1742-6596/1934/1/012012).

See previous answer.

Minor revisions (require changes in the text):

- 1. line 150- The placement of the farm parameterization in the cell extrusion buffer zone can cause problems both on the Gaussian distribution of the drag force as well as in the resolution of the wind farm wakes. Please discuss this in the manuscript. The magenta area as shown in Fig. 2 only has a limited stretching from D/8 up to a maximum cell size of 2D. The grid refinement study from Section 4.1.2 showed that this maximum cell size is sufficient and therefore we do not see the problem that you describe. Note that this is already explained in Section 3.1. It could that the amount of cell stretching as depicted in Fig. 2 had been misunderstood as we only show every 32nd cell (as described in the caption).
- 2. line 160- I would like you to explain the meaning of "inlet boundary condition" applied to the top. Good point. Using an inlet on top for the wind farm (cluster) simulations means that we have a liddriven setup, where the values of the wind speed components and turbulence quantities are set based on a 1D numerical precursor simulation. For the precursor simulation, we used a symmetry condition on top and a prescribed pressure gradient to drive the flow. We added ...hence the flow is lid-driven.
- 3. line 180- Why do you use generic Ct and Cp values for the turbine installed in Dudgeon? The thrust coefficient curves of the SGRE wind turbines are proprietary. Therefore, we have chosen to use a generic turbine for the verification study such that another researcher can more easily redo our work.
- 4. line 180- Could you review the description of the AD model used? Different AD models are mentioned in the text and it is not clear to the reader what you finally use. We believe that the text in Section 3.1.1 already clearly states which AD model setup is used in either the validation or the verification study.
- 5. line 185- Have you verified if the resolution of the mesh on the disk (D/8) is sufficient for applying non-uniform thrust and tangential forces? Yes, we have applied a grid refinement study for non-uniform thrust and tangential forces in a previous work (van der Laan et al., 2015b), which showed that a resolution of D/8 is sufficient. We have added this reference to the text.
- 6. line 195- The use of a constant Ct throughout the farm is a very strong simplification, especially when simulating velocities close to or higher than the rated velocity. At these speeds the Ct of the first row of turbines is lower than the turbines impacted by wakes and therefore with lower velocity. Please clarify this limitation in the text.

You are right about this. However, the main application of the AWF model is to be an obstacle for another wind farm of interest represented by ADs (RANS-AD-AWF). In that case, it is mainly the total wind farm drag force that is driving the wind farm wake of the AWF model and the  $C_T$  distribution in the wind farm is less important. The secondary application of the AWF model is to be like a 'sensor' of wind farm power, by modeling all wind farms in wind farm cluster by AWF models (RANS-AWF). In this case, the  $C_T$  distribution could indeed become important. The use of variable  $C_T$  and more local AWF  $C_T$  controller is planned to be future work. In addition, the changes in article text are provided in the answer to comment 10.

7. line 200- The justification for why you don't inject extra TKE is a bit vague. Could you justify in a more complete way?

Adding TKE sources opens up a whole new study because there are many different types of TKE sources in the literature. It is not even clear if one should either add (based on dispersive stresses) or remove TKE by adding source of TKE dissipation in the  $\varepsilon$  equation (based on forest canopy modeling from Sogachev et al. (2012)). We believe that it is better to investigate the use of TKE source for the AWF wind farm model in a dedicated follow-up article. We have added/modified the following text in Section 3.1.2:

One could employ an additional term in the turbulence model equations to account for the effect of under-resolving the wind farm layout in the AWF model when using large horizontal cells (Abkar and Porté-Agel, 2015). However, the literature is divided about whether this extra term should be zero (Volker et al., 2015), act as a source (Fitch et al., 2012; Abkar and Porté-Agel, 2015) or sink of turbulence kinetic energy (Sogachev et al., 2012). Investigating this is not in the scope of this study and so we do not use a source term of turbulence, partly because we already get reasonable results with only a momentum sink (Eq. 1).

- 8. line 215- You propose to distribute the drag force in a Gaussian way. Have you checked if farm size and equivalent wake spanwise width is affected by this strategy? Yes, because we have investigated two different farm sizes in the verification study by using a wind turbine spacing of 4D and 8D (Figs. 9 and 10, now Figs. 10 and 11). We still get reasonable results for the higher density wind farm with respect to RANS-AD. When the standard deviation is increased in the Gaussian method then the wind farm wake becomes more smooth as shown in Figs. 6 and 7 (now Figs. 7 and 8.
- 9. line 235- One of the main limitations of the farm parameterization that you propose is the need to use pre-calculated wind farm drag force magnitude, which depends on both velocity and direction. This brings with it a large computational cost of simulating each farm with AD for different inflow conditions. Even though you mention this in line 580, in the abstract you do not take into account this extra cost in the percentages. In my opinion, this makes the use of the parameterization that you propose very impractical. I know this is your first approach to the farm parameterization in RANS, but in future work I would like to see more developments on the model.

You are right that the required input for the AWF model (wind farm thrust and power) calculated by RANS-AD wind farm simulations is an expensive step. We are currently working on a method to calculate these inputs with a RANS-based surrogate model approach. The first results look promising and we plan to present this work at the Wake Conference, 2023. We have added a reference to this work:

An initial study shows that is possible to predict  $C_{T,wf}$  and  $C_{P,wf}$  within a mean absolute error of around 1% when using a RANS-based surrogate wind farm model (van der Laan et al., 2023).

10. line 260- it is not clear if the parameterization can respond to a non-uniform inflow condition (such as the partial wake impact of another upstream farm). Looks like in line 530 you found the problem related to this. Please clarify this in the text.

Good point, we have clarified this in Section 3.1.2:

In steps 1 and 2, we neglect the influence of inhomogeneous inflow conditions on the wind farm thrust and power, as for example partial wind farm wake effects of the neighboring wind farms. However, the AWF model (as applied in Step 2 with a force controller) can partly respond to inhomogeneous conditions because the local thrust forces dependent on the local velocity, although variations in wind turbine thrust coefficients are not captured due to the use of a global wind farm thrust coefficient. The impact of this assumption is investigated in Sect. 4.2.1. We have also adapted Sect 4.2.1: When the downstream wind farm is operating in a half wind farm wake situation, mostly applicable for the wind directions 220-225 and  $245-250^{\circ}$ , then the entire AWF is still affected if the change in the AWF volume averaged wind speed or wind direction changes the wind farm thrust coefficient, which would not be the case when the downstream wind farm is represented by ADs.

11. line 370- Why do you use the explicit wake parameterization (EWP) instead of the Fitch parameterization? Especially when a good comparison has been found between RANS and Fitch (https://wes.copernicus.org/articles/7/1069/2022/)

You are right that we could have chosen to employ the Fitch wind farm parametrization scheme or other schemes. The work that you cite indeed showed that the Fitch scheme performed better than the EWP scheme for one particular double-wind farm case. A general conclusion cannot be made based on this work, as the performance of the employed wind farm parametrization may be case-dependent. The WRF simulations of the currently investigated wind farm cluster including the EWP parametrization were already performed prior to the development of the AWF model. Future work could compare both parametrizations with the RANS-AWF model.

Thank you for your good manuscript and I hope you find my comments adequate and useful to improve the quality of the work.

**Reviewer 2**

This paper propsed a wind farm model which works for RANS with relatively large CFD cell sizes. The model uses tabulated wind farm thrust coefficient based on precursor RANS-AD simulations and the Gaussian-function force distribution. Also, a new atmospheric inflow condition is proposed for the neutral condition based on analytical potential temperature profile. The analysis is comprehensive and the results look promising. The save in the computation time is significant. I believe this work is very good and meaningful, and, therefore, recommend acceptance with minor revisions, such as:

- 1. The abstract is too long, where SCADA is used without definition. The abstract is indeed quite lengthy. We have removed the text about TurbOPark in the abstract, as this is a secondary result, and we have shortened the text about the new inflow model. In addition, the definition of SCADA has been added to the introduction and removed in the abstract.
- 2. Line 186 and 187, the brackets of the citations are missing. Adopted.
- 3. I understand that there are two sets of meshes, i.e., one is for CFD, and the other for the AWF model. But it is a little confusing when I was reading the paper. It would be better to make it more clear, such as adding a figure showing both the meshes and their relationship. This is good suggestion. We have added a sketch and description of the three different domain types in Section 3.1:

In this work, different RANS flow domains are used to model single, double and triple wind farms. A sketch of the flow domain types of the applied methods is depicted in Fig. 2. All flow domains are Cartesian grids where the inflow direction at the reference height is aligned with x-axis and different wind directions are modeled by rotating the wind farm cluster around (x, y) = (0, 0), while maintaining the grid and the global inflow direction. The RANS-AD method (Fig. 2a) represents all wind turbines in a wind farm by ADs. Each wind turbine is treated as a single model, which has its own polar grid that is connected to the flow domain, and is also controlled individually. The cells around the ADs, as marked by the cyan box in Fig. 2a, are uniformly spaced in the horizontal dimensions using a fine resolution in order to resolve the wind turbine wakes. The RANS-AWF method (Fig. 2b) follows a similar flow domain as the RANS-AD method; however, each wind farm is considered as a single model that uses its own force controller. The wind turbine forces are distributed in a Cartesian grid that encapsulates the entire wind farm and this Cartesian grid is then connected to the flow domain grid. The refined area in the RANS-AWF method can be an order of magnitude coarser compared the refined area for the RANS-AD model, which is investigated in detail in Sect. 4.1. The third domain type is depicted in Fig. 2c and represents the RANS-AD-AWF method, where both AD and AWF models are present. Since the AWF model does not require the fine spacing of the AD models, a second region in the flow domain is defined, as marked by the magenta box in Fig. 2c, where the cells are expanded in the horizontal direction while moving away from the cyan box, up to a maximum set spacing. It should be noted that the three methods as depicted in Fig. 2 can also be applied to simulate multiple wind farms. In this case, the size of the refined inner domain(s) are adjusted to resolve all wind turbine and farm wakes. An overview of the all the applied domains for the validation cases, as well as the computational effort is provided in Tab. 3.

4. Around line 255, how are those RANS-AD simulations set up, such as domain size, grid resolution, boundary conditions, etc? Are they the same as the later RANS-AWF clustered case but only consider one wind farm at time? In the conclusion, it says a 80% saving even if RANS-AD is used as the precursor for a case consist of there wind farms. I assume for each wind farm, you have to run the RANS-AD for a series of wind speed and directions, then how can the combination of all those effort save anything from running a single RANS-AD case with all the three wind farms together? I suppose the precursor RANS-AD is way cheaper, but the manuscript is not clear about it. Please clarify. The domain setup of the RANS-AD method has been clarified as discussed in the previous answer. In addition, we have added a more clear Table of the meshes and computational effort of the validation cases:

The reason why three single wind farm simulation with RANS-AD (to calculate  $C_{T,\text{wf}}$  and  $C_{P,\text{wf}}$ ) is cheaper compared to the directly using RANS-AD for the entire cluster has to do with the large fine spaced inner grid needed to resolve all three wind farms. The number of cells in total for the three single wind farms is 115+163+115=393 million while the grid covering all wind farms is 1193 million. If the distance between the wind farms would be less, then it could be that employing the RANS-AD method for the entire cluster is as expensive as simulating the three single wind farms separately as a precursor step.

**Reviewer 3**

The manuscript presents a new inflow generation method for RANS and the introduction of Actuator Wind Farm parametrization. The study is novel and addresses aspects of high relevance to the community. The work is very extensive, and the manuscript is overall very well-written. The work is of high interest of readers of Wind Energy Sciences, and I recommend publication subject to changes/corrections/clarifications on the points indicated below.

1. The way the specific CPU time reductions are mentioned in the abstract is somewhat misleading, as the step to perform RANS-AD simulations for the parametrization of the WIND farm curves is not mentioned while it has significant computational overhead. We have clarified this with a more detailed Table 3 as shown for Answer 4 to Reviewer 2. We have

| Case                                       | Method      | Cells Block Blocks Wall cloc |      | Wall clock | CPU hours   CPU hours inc |               |                 |  |
|--------------------------------------------|-------------|------------------------------|------|------------|---------------------------|---------------|-----------------|--|
|                                            |             | [million]                    | size | & CPUs     |                           |               | Steps 1a and 1b |  |
| Single AWF $C_{T, wf}$                     |             |                              |      |            |                           |               |                 |  |
| ShS                                        | RANS-AD     | 115                          | 64   | 437        | 0.20                      | 88            | -               |  |
| Rb                                         | RANS-AD     | 163                          | 64   | 621        | 0.25                      | 153           | -               |  |
| Du                                         | RANS-AD     | 115                          | 64   | 437        | 0.22                      | 102           | -               |  |
| Single AWF force                           |             |                              |      |            |                           |               |                 |  |
| Shs                                        | RANS-AWF    | 1.05                         | 32   | 32         | 0.0080                    | 0.25          | -               |  |
| Rb                                         | RANS-AWF    | 1.31                         | 32   | 40         | 0.0072                    | 0.28          | -               |  |
| Du                                         | RANS-AWF    | 1.31                         | 32   | 40         | 0.0066                    | 0.26          | -               |  |
| Double wind farm                           |             |                              |      |            |                           |               |                 |  |
| Du + ShS                                   | RANS-AD     | 297                          | 64   | 1134       | 0.72                      | 819           | -               |  |
| Du + ShS                                   | RANS-AD-AWF | 110 (-63.0%)                 | 64   | 420        | 0.50 (-32.6%)             | 204 (-75.1%)  | 385~(-53.0%)    |  |
| Du + ShS                                   | RANS-AWF    | 3.15 (-98.9%)                | 32   | 96         | 0.027 (-96.7%)            | 2.30 (-99.7%) | 421 (-48.6%)    |  |
| Wind farm cluster                          |             |                              |      |            |                           |               |                 |  |
| Du + ShS + RB                              | RANS-AD     | 1193                         | 64   | 4550       | 1.53                      | 6958          | -               |  |
| $\mathrm{Du} + \mathrm{ShS} + \mathrm{RB}$ | RANS-AD-AWF | 252 (-78.9%)                 | 64   | 960        | 0.56 (-63.4%)             | 538 (-92.3%)  | 1282 (-80.1%)   |  |
| $\mathrm{Du} + \mathrm{ShS} + \mathrm{RB}$ | RANS-AWF    | 6.55 (-99.5%)                | 32   | 200        | 0.026 (-98.3%)            | 2.23 (-99.9%) | 1206 (-82.7%)   |  |

Table 4: Grid size and computational effort of RANS wind farm (cluster) simulations including Dudgeon (Du), Sheringham Shoal (ShS) and Race Bank (RB). Percentage between brackets reflects the reduction when using AWF models with respect to only using ADs. CPU and run hours are listed per flow case.

also added a sentence in the abstract to clarify:

If the wind farm thrust and power coefficients inputs are derived from RANS-AD simulations then the CPU-time reduction is still 82.7% for the wind farm cluster case.

- 2. Line 259-267: The computational overhead of the RANS-AD simulations is not quite clear. I am unsure what you want to convey with the statement, "The computational costs of step 1a could be alleviated ...". The AWF simulation should always have significant benefits over the AD simulations. If it were possible to run AD simulations at a very limited computational cost, the AWF simulation approach would not be required. Improving the entire modeling suit would not necessarily reduce the overhead of step 1a. Of course, it would be interesting to know the effect of the engineering models, which is a way to reduce the computational time of step 1a, on the accuracy. This is not discussed here, but given that the work is already very comprehensive, it is reasonable to leave this for future work. We believe this has been clarified with the more comprehensive table as discussed in the previous answer.
- 3. Line 635: Much clearer than in the abstract. Still, specific numbers are case-dependent. See Answer 4 to Reviewer 2.
- Line 465-475: A more detailed explanation of the different grids employed would be helpful here. This discussion is difficult to follow here. See Answer 3 to Reviewer 2.
- 5. Line 487: Figure 11-14: For completeness, can you define how the Gaussian filter of 5 degrees is defined?

This is a weighted average of the results obtained from different wind directions, where the weights are defined by a Gaussian function normalized with the numerical integral of the same Gaussian function such that the sum of the weights is equal to one. The use of Gaussian filter in comparisons with SCADA is very common but indeed rarely mathematically described, although one can find such a description in Antonini et al. (2019), which we have now added as a reference in Section 4.2.1.

6. Figure 13/14: An explanation of how the flow visualizations in figure 13/14 should be interpreted would be appreciated. Is the choice of visualization direction related to the simulation approach? An explanation of this would be helpful.

The reviewer is perhaps used to working with plots where the North is always direction to the top

and this may lead to some confusion when looking at our rotated contour plots. We always set a wind direction of 270° when working with RANS wind farm simulations for flat, homogeneous terrain and we model the effect of the wind direction by rotating the wind farm layout, as described in Section 3.1. This provides the advantage that we can run a set of wind direction flow cases consecutively in a single simulation. Once the first wind direction case has converged we save the results, rotate the wind farms/turbines positions and then continue the simulation. This method reduces the total number of required iterations necessary to convergence the flow as discussed in previous work (van der Laan et al. (2019, 2022)) because only local changes need to recalculated. We have rotated the flow in Figs. 13 and 14 to better visualize the incoming flow for the front rows of the Dudgeon wind farm for which a comparison with the SCADA measurements have been made. We have added the following clarification in the text for Fig. 11 (which is first time we introduce this type of plot):

The contours plots in Figs. 11a-e are rotated to better visualize the incoming flow for the front row wind turbines and transect, for which the results are depicted in Figs. 11f-h.

Another "fun feature" of Figs. 13 and 14 is that one can scroll through the pdf document and see the sub figures as snap shots of an animation. (We could consider to actually make such an animation and provide this as an additional data object if the article is accepted for publication in WES.)

Typos:

- 1. Line 110: Both wind turbine types  $\rightarrow$  this is confusing as it is only printed in the table. We have removed *types* from the text and Table 1. (The phrase *wind turbine type* is common to PyWake software users but maybe not widely used in the wind energy community.)
- 2. Line 152: Define growth ratio The cell growth ratio is defined as  $\delta_{i+1}/\delta_i$  where  $\delta_i$  is the cell size in direction *i* and  $\delta_{i+1}$  is the cell of a neighboring cell orientated in the expansion direction. We have written *expansion ratio* instead of *growth ratio*, as the first is more commonly used in CFD.
- 3. Line 164-172: This text is a bit difficult to follow; it is much clearer in table 3. However, that is only presented much later. We believe that the addition of the figure shown in Answer 3 to Reviewer 2 could help understanding the RANS-AD-AWF type that is explained in detail in this part.
- 4. Line 192: Please clarify what you mean by "and the provided thrust coefficients". We have extended this sentence: ...and the provided thrust coefficient curves of the SGRE wind turbines are employed.
- 5. Line 679: MMP is a typo? This is an abbreviation of Mads Mølgaard Pedersen, who initially was a coauthor but later requested to be named in the acknowledgments instead. We have now removed MMP from the author contributions section.

**Own** improvements**

- 1. A number of small fixes have been applied to the AWF model implementation in PyWakeEllipSys and has lead to a slightly better match between the RANS-AD and RANS-AWF in Fig. 12 (Fig. 13 in revised article). In addition, the inner grid dimensions (as marked with the cyan area in Fig. 2, now Fig. 3 in the revised article) has been reduced following a better description of the possible wind farm orientations (that are used to model different wind directions). This has resulted in reduced grid sizes and small changes for the validation cases.
- 2. The TurbOPark implementation has been adapted to better match the setup from Ørsted, which has lead to stronger wake effects in Fig. 12 (previously Fig. 11). These changes are:
  - Update of the rotor average model from GQGridRotorAvg(4, 3) to GaussianOverlapAvgModel.
  - use\_effective\_ws=True to use\_effective\_ws=False .

- Removal of turbulence model as this was already switched off by using use\_effective\_ti=False.
- 3. A number of typos are fixed in Table 3 regarding the number of blocks for the RANS-AD-AWF method for the wind farm cluster case and the CPU hours for the RANS-AD Double wind farm case.

**Appendix B: Comparison and challenges of RANS inflow models applied to the wind farm cluster validation case**

A new atmospheric inflow model is proposed in Sect. 3.1.3 and used to perform the RANS wind farm cluster simulations in Sect. 4 The new inflow model employs a prescribed temperature profile with an inversion that sets the ABL height. Previous work employed an ABL inflow model based on a global turbulence length scale limiter,  $\ell_{\text{max}}$  (Apsley and Castro, 1997; van der Laan and Sørensen, 2017; van der Laan et al., 2017). For a low value of  $\ell_{\text{max}}$ , a lower ABL height is obtained. A major issue with the global turbulence length scale limiter is that it can also limit the wind farm wake turbulence length scales that could lead to a nonphysical slow wake recovery. This issue was mitigated by switching off the global length scale limiter in regions where high velocity gradients are present (van der Laan and Sørensen, 2017), although it is unclear if this ad-hoc solution is sufficient for wind farm cluster simulations. Another challenge is that for low ABL heights, one can obtain numerical oscillations, which both the new model and the model based on  $\ell_{\text{max}}$  can suffer from. This section is meant to illustrate these issues and to show the difference between the inflow models when applied to the wind farm cluster validation case.

|             |                  | Input parameters |       |            |                                         |                   | Derived parameters |                      |                       |                  |
|-------------|------------------|------------------|-------|------------|-----------------------------------------|-------------------|--------------------|----------------------|-----------------------|------------------|
|             |                  | $I_{\rm amb}$    | $z_i$ | $\theta_0$ | $\frac{\partial \theta}{\partial z} _c$ | $\frac{z_T}{z_i}$ | $z_0$              | G                    | $z_0$                 | $\ell_{\rm max}$ |
| Case        | Model            | [-]              | [m]   | [K]        | [K m -1 ]                    | [-]               | [m]                | [m s -1 ] | [m]                   | [m]              |
| Tall ABL    | Prescribed temp. | $10^{-5}$        | 1000  | 285        | $5 \times 10^{-3}$                      | 0.2               | -                  | 8.50                 | $3.25 \times 10^{-5}$ | -                |
| Tall ABL    | $\ell_{ m max}$  | $10^{-6}$        | -     | -          | -                                       | -                 | $5 \times 10^{-5}$ | 8.62                 | -                     | 53.3             |
| Shallow ABL | Prescribed temp. | $10^{-5}$        | 300   | 285        | $5 	imes 10^{-3}$                       | 0.4               | -                  | 8.54                 | $1.77 	imes 10^{-4}$  | -                |
| Shallow ABL | $\ell_{ m max}$  | $10^{-6}$        | -     | -          | -                                       | -                 | $5 \times 10^{-3}$ | 8.93                 | -                     | 7.62             |

Table 4: Summary of input and derived parameters for the ABL inflow models, both using  $I_{\text{ref}} = 0.044$ ,  $U_{\text{ref}} = 8 \text{ m s}^{-1}$ ,  $z_{\text{ref}} = 102 \text{ m and } f_c = 1.168 \times 10^{-4} \text{ s}^{-1}$ .

Table 4 lists the input and derived parameters of a shallow and a tall ABL inflow using the new prescribed temperature ABL model and the ABL model based on  $\ell_{\rm max}$ . The tall ABL case is the same as used through out the article (Sects. 3.1.3 and 4), while the shallow ABL is an additional case meant to illustrate the challenges with the two inflow models. For the  $\ell_{\max}$  ABL inflow model, the derived values for G and  $\ell_{\max}$ are found by interpolating a pre-calculated library of ABL profiles (van der Laan et al., 2020) for given  $z_0, f_c, I_{ref}$  and  $U_{ref}$  values. The prescribed temperature model uses an optimizer for G and  $z_0$  instead, as discussed in Sect. 3.1.3 The tall ABL profile from the  $\ell_{max}$  ABL inflow model uses a similar roughness length  $(z_0 = 5 \times 10^{-5} \text{ m})$  as was calculated for the inflow model based on the prescribed temperature (Tab. 2), which results in a relative large value of  $\ell_{\rm max}$ , namely 53.3 m. The shallow ABL profile of the prescribed temperature ABL model is generated with a lower  $z_i$  (300 m) resulting in a larger derived  $z_0$  (Tab. 4). In addition, the value of  $z_T/z_i$  as used in the prescribed temperature ABL model is increased for the shallow ABL to avoid an inflection of the wind speed profile below the super geostrophic jet. To lower the ABL height for model based on  $\ell_{\text{max}}$ , one can lower  $z_0$  or  $I_{\text{ref}}$ ; we choose to set  $z_0 = 5 \times 10^{-3}$  m. This results in a derived value of  $\ell_{\rm max}$  of only 7.62 m. In addition, the value of  $z_T/z_i$  as used in the prescribed temperature ABL model is increased for the shallow ABL to better match the shallow ABL profile of the  $\ell_{\rm max}$  model. Finally, the ABL model based on  $\ell_{\rm max}$  is employed with a lower ambient turbulence intensity value,  $I_{\rm amb}$ , compared to the prescribed temperature model because the  $\ell_{max}$  ABL model is more sensitive to this parameter. The chosen values for each ABL model are sufficient to not influence the numerical solution (van der Laan et al., 2020). The results of all profiles are depicted in Fig. 1. Here it is clear that over the rotor swept area, the profiles of wind speed, direction, and normalized turbulence intensity resulting from the prescribed temperature model are similar to those resulting from the  $\ell_{\rm max}$  model for the tall ABL case. The main difference can be found in the turbulence model length scale (Fig. 1d), where the prescribed temperature ABL model predicts larger

length scales over the rotor swept area compared to the  $\ell_{\rm max}$  based ABL model. For taller altitudes, the models produce quite different ABL solutions probably because the prescribed temperature model sets an explicit inversion strength while the ABL model based on  $\ell_{\rm max}$  calculates this implicitly. For the shallow ABL case, both models predict very different profiles. This is because the model based on  $\ell_{\rm max}$  represents stable conditions for low values of  $\ell_{\rm max}$  while the prescribed temperature ABL model reflects a conventionally neutral ABL.